

# Accounting for the effects of surface BRDF on satellite cloud and trace-gas retrievals: A new approach based on geometry-dependent Lambertian-equivalent reflectivity applied to OMI algorithms

A. Vasilkov[1], W. Qin[1], N. Krotkov[2], L. Lamsal[3], R. Spurr[4], D. Haffner[1], J. Joiner[2], E.-S. Yang[1], and S. Marchenko[1]

[1]Science Systems and Applications Inc., Lanham, MD, USA
[2]NASA Goddard Space Flight Center, Greenbelt, MD, USA
[3]Universities Space Research Association, Columbia, MD, USA
[4]RT Solutions, Cambridge, MA, USA

*Correspondence to:* A. Vasilkov
(alexander.vasilkov@ssaihq.com)

**Abstract.** Most satellite nadir ultraviolet and visible cloud, aerosol, and trace-gas algorithms make use of climatological surface reflectivity databases. For example, cloud and $NO_2$ retrievals for the Ozone Monitoring Instrument (OMI) use monthly gridded surface reflectivity climatologies that do not depend upon the observation geometry. In reality, reflection of incoming direct and diffuse so-

lar light from land or ocean surfaces is sensitive to the sun-sensor geometry. This dependence is described by the bidirectional reflectance distribution function (BRDF). To account for the BRDF, we propose to use a new concept of geometry-dependent Lambertian-equivalent reflectivity (LER). Implementation within the existing OMI cloud and $NO_2$ retrieval infrastructure requires changes only to the input surface reflectivity database. The geometry-dependent LER is calculated using a

vector radiative transfer model with high spatial resolution BRDF information from the MODerate-resolution Imaging Spectroradiometer (MODIS) over land and the Cox-Munk slope distribution over ocean with a contribution from water-leaving radiance. We compare the geometry-dependent and climatological LERs for two wavelengths, 354 and 466 nm, that are used in OMI cloud algorithms to derive cloud fractions. A detailed comparison of the cloud fractions and pressures derived with

climatological and geometry-dependent LERs is carried out. Geometry-dependent LER and corresponding retrieved cloud products are then used as inputs to our OMI $NO_2$ algorithm. We find that the use of high-resolution geometry-dependent LERs can increase $NO_2$ vertical columns by up to 50% in highly polluted areas.



## 1 Introduction

Satellite ultraviolet and visible (UV/Vis) nadir backscattered sunlight trace-gas, aerosol, and cloud retrieval algorithms must accurately estimate the reflection by the Earth's surface in order to produce high quality data sets. Surface reflectivity climatologies used in most current algorithms are typically gridded monthly Lambertian-equivalent reflectivities (LERs) that have been derived from satellite observations (e.g., Herman and Celarier, 1997; Kleipool et al., 2008; Russell et al., 2011).

These climatologies generally have no dependence on the observation geometry. However, it is well known that both ocean and land reflectivity depend upon viewing and illumination geometry. This dependence is described by the bidirectional reflectance distribution function (BRDF), mathematically expressed as

$$\mathrm{BRDF}(\omega_i, \omega_r) = \frac{dI(\omega_r)}{I(\omega_i)cos(\theta_i)d\omega_i}, \tag{1}$$

where $dI(\omega_r)$ is the portion of total radiance reflected in the direction defined by the vector $\omega_r$ due to the illuminating irradiance, $F$, from the direction defined by the vector $\omega_i$: $dF(\omega_i) = I(\omega_i)cos(\theta_i)d\omega_i$, $\theta_i$ is the angle between the normal to the surface and the direction of illuminating light, and $d\omega_i$ is the element of the solid angle (Nicodemus, 1965; Schaepman-Strub et al., 2006; Martonchik et al., 2000). When the surface is illuminated by a parallel beam of light, the integral

over the solid angle of reflected light

$$\mathrm{BSA}(\omega_i) = \int \mathrm{BRDF}(\omega_r, \omega_i) \cos(\theta_r) d\omega_r \tag{2}$$

provides the so-called Black Sky Albedo (BSA) of the surface. It follows from Eq. 2 that the BRDF of a perfect Lambertian surface is equal to $1/\pi$.

The frequently used dimensionless bidirectional reflectance factor (BRF) is defined as "the ratio

of the radiant flux reflected by a sample surface to the radiant flux reflected into the identical beam geometry by an ideal (lossless) and diffuse (Lambertian) standard surface, irradiated under the same conditions as the sample surface" (Schaepman-Strub et al., 2006). In general, the relationship between BRF and BRDF for an arbitrary surface can be obtained from Eq. 1 by using BRDF=$1/\pi$ for an ideal Lambertian surface, i.e.,

$$\mathrm{BRF}(\omega_i, \omega_r) = dI(\omega_i, \omega_r)/dI_{Lam}(\omega_i) = \pi\mathrm{BRDF}(\omega_i, \omega_r). \tag{3}$$

BRF and BRDF are both inherent properties of the surface that do not depend on the illumination conditions (Schaepman-Strub et al., 2006). While BRDF is a function describing a surface for all possible illuminating and reflected directions, the BRF refers to a specific illumination and observational geometry for a given measurement. BRF from satellite observations can therefore differ

significantly for the same area over different days owing to variations in sun-satellite geometries. In other words, for a given surface BRDF is always the same (neglecting seasonal changes), but BRF changes from day to day depending on observational conditions.



Many satellite UV/Vis algorithms are based on the so-called mixed Lambert-equivalent reflectivity (MLER) model, first introduced by Seftor et al. (1994). For example, the MLER concept is currently used in most trace gas (Boersma et al., 2011; Bucsela et al., 2013) and cloud (Acarreta et al., 2004; Joiner and Vasilkov, 2006) retrieval algorithms for the Ozone Monitoring Instrument (OMI), a Dutch/Finnish UV/Vis sensor (Levelt et al., 2006) flying on the NASA Aura satellite. The MLER model treats cloud and ground as horizontally homogeneous Lambertian surfaces and mixes them using the independent pixel approximation (IPA). According to the IPA, the measured top-of-atmosphere (TOA) radiance is a sum of the clear sky and overcast subpixel radiances that are weighted with an effective cloud fraction (ECF). The ECF is calculated by inverting

$$I_m = I_g(R_g)(1 - \text{ECF}) + I_c(R_c)\text{ECF} \qquad (4)$$

at a wavelength not substantially affected by rotational-Raman scattering (RRS) or atmospheric absorption, where $I_m$ is the measured TOA radiance, $I_g$ and $I_c$ are the precomputed clear sky (ground) and overcast (cloudy) subpixel radiances, $R_g$ and $R_c$ and the corresponding ground and cloud Lambertian-equivalent reflectivities (LERs), respectively.

The MLER model typically assumes $R_c = 0.8$. This value of $R_c$ was used by McPeters et al. (1996) for a UV total column $O_3$ algorithm and independently derived by Koelemeijer et al. (2001) for use in near-infrared $O_2$ A-band cloud pressure retrievals. The assumption of $R_c = 0.8$ effectively accounts for Rayleigh scattering in partially cloudy scenes (Ahmad et al., 2004). This approach also accounts for scattering/absorption that occurs below a thin cloud.

The MLER model compensates for photon transport within a cloud by placing the Lambertian surface somewhere in the middle of the cloud instead of at the top (Vasilkov et al., 2008). As clouds are vertically inhomogeneous, the pressure of this surface does not necessarily correspond to the geometrical center of the cloud, but rather to the so-called optical centroid pressure (OCP) (Vasilkov et al., 2008; Joiner et al., 2012). The cloud OCP can be thought of and modeled as a reflectance-averaged pressure level reached by back-scattered photons (Joiner et al., 2012). Cloud OCPs are the appropriate quantity for use in trace-gas retrievals from similar instruments (Vasilkov et al., 2004; Joiner et al., 2006, 2009).

In one of the few studies to explore the effects of surface BRDF on satellite trace gas retrievals, Zhou et al. (2010) show how various treatments of surface reflectance, including BRDF, affect OMI tropospheric $NO_2$ retrievals over Europe. Their study, which covers the months of July and November, suggested that account of surface BRDF effects can change $NO_2$ retrievals by up to 20% with the largest effects at high view angles. Ignoring the surface BRDF can also introduce $NO_2$ retrieval errors that vary with land type (Noguchi et al., 2014). In an effort to improve $NO_2$ retrievals over China, Lin et al. (2014, 2015) revised the calculation of tropospheric air mass factor (AMF) in the Dutch OMI $NO_2$ (DOMINO) product using improved information for cloud, aerosols, and BRDF from the MODerate-resolution Imaging Spectroradiometer (MODIS); they reported better agreement with independent $NO_2$ observations. Our motivations for this work follow from these studies





that offered valuable insights into the effects of the surface BRDF on $NO_2$ retrievals. We continue
in this line of investigation by (1) examining in detail the BRDF effect on retrieved cloud parame-
ters that are important inputs for trace-gas retrievals including $NO_2$; (2) additionally investigating
BRDF impact on cloud and $NO_2$ retrievals over ocean; and (3) providing a computationally efficient
method of accounting for BRDF effects that can be incorporated into existing retrieval algorithms
with minimal changes.

To account for surface BRDF, we introduce the concept of a geometry-dependent surface LER.
The geometry-dependent LER is derived from TOA radiance computed with Rayleigh scattering
and BRDF for the particular geometry of a satellite-based pixel. This approach does not require any
major changes to existing MLER trace gas and cloud algorithms. The main revision to the algorithms
requires replacement of the existing static LER climatologies with LERs calculated for specific field-
of-view (FOV) sun-satellite geometries. The geometry-dependent surface LER approach can be
applied to any current and future satellite algorithms that use the MLER concept.

We implement the geometry-dependent LERs based on a MODIS BRDF product and use these
LERs within OMI cloud and $NO_2$ algorithms. It should be noted that the MODIS BRDF product is
derived from the atmospherically corrected TOA reflectances (i.e., aerosol and Rayleigh scattering
effects are removed at the high spatial resolution of MODIS). In contrast, the climatological LERs
currently used in OMI algorithms, from either the Total Ozone Mapping Spectrometer (TOMS) or
OMI, are derived by correcting only for Rayleigh scattering and thus include aerosol effects (see
details in Herman and Celarier, 1997; Kleipool et al., 2008). Therefore, the use of the geometry-
dependent LER product in trace gas algorithms over heavily polluted regions may also require an
explicit account of aerosols (Lin et al., 2015). In this study we do not consider aerosol effects.

## 2  Satellite data sets and radiative transfer model

### 2.1  VLIDORT code

For all radiative transfer (RT) calculations, we use the Vector Linearized Discrete Ordinate Radiative
Transfer (VLIDORT) code (Spurr, 2006). VLIDORT computes the Stokes vector in a plane-parallel
atmosphere with a non-Lambertian underlying surface. It has the ability to deal with attenuation
of solar and line-of-sight paths in a spherical atmosphere, which is important for large solar zenith
angles (SZA) and viewing zenith angles (VZA). Unlike Lin et al. (2014, 2015), we use a vector code
because neglect of polarization can lead to considerable errors for modeling backscatter spectra
in UV/Vis. This is particularly the case for modeling backscatter spectra over the ocean where
reflection of unpolarized light from the flat ocean surface at the Brewster angle leads to its perfect
linear polarization (Vasilkov et al., 1990a,b).





## 2.2 MODIS BRDF data set

We use the MODIS gap-filled BRDF Collection 5 product MCD43GF (Schaaf et al., 2002, 2011).
This product provides three coefficients, $a_i$, as a function of time and spatial coordinates for three
BRDF kernels: 1) isotropic, $k_{iso} \equiv 1$; 2) volumetric, $k_{vol}$; and 3) geometric, $k_{geo}$. The BRDF
coefficients are dynamic, i.e., 16-day averages for every 8 days of the year from 2003 to present.
They are provided for snow-free land and permanent ice at a high spatial resolution (30 arc sec). In
this study we do not consider temporary snow-covered areas. In principal, those areas can be treated
with the approach of McLinden et al. (2014) that is based on the MODIS-derived albedo product.
Unlike Lin et al. (2015), we do not use MODIS data over coastal zones and inland waters, because
the MODIS kernel model is not applicable for water surfaces. Instead of MODIS data, we apply our
ocean model described in Section 3 to the coastal zones and inland waters.

## 2.3 OMI data sets

### 2.3.1 OMI cloud algorithms

In this paper, we examine the BRDF effects on two OMI cloud algorithms, one based on rotational-
Raman scattering (RRS) in the UV and the other on $O_2$-$O_2$ absorption at 477 nm. The $O_2$-$O_2$ cloud
algorithm developed by the authors and used here is similar to an operational $O_2$-$O_2$ cloud algorithm
developed at the Royal Meteorological Institute of the Netherlands (KNMI), known as OMCLDO2,
(Acarreta et al., 2004; Sneep et al., 2008), but differs in a few respects described below.

Both the RRS and $O_2$-$O_2$ algorithms utilize the MLER concept. We use 354 and 466 nm in the
RRS $O_2$-$O_2$ algorithms, respectively, to compute ECF. It should be noted that the ECF implicitly
accounts for non-absorbing aerosols, treating them as clouds and this increases cloud fraction.

The OMI RRS cloud algorithm is detailed in Joiner et al. (2004), Joiner and Vasilkov (2006), and
Vasilkov et al. (2008). OCP is derived from the high-frequency structure in the TOA reflectance
caused by RRS in the atmosphere. The OCP is retrieved by a minimum-variance technique that
spectrally fits the observed TOA reflectance within the spectral window of 345.5–354.5 nm. The
RRS algorithm does not report the cloud OCP for ECF< 0.05 owing to large retrieval errors at small
values of ECF (Vasilkov et al., 2008).

Our $O_2$-$O_2$ cloud algorithm retrieves OCP from OMI-derived oxygen dimer slant column densities
(SCD) at 477 nm. Our algorithm spectral fitting differs from KNMI's in that it utilizes temperature-
dependent $O_2$-$O_2$ cross-sections (Thalman and Volkamer, 2013) and incorporates a new fitting tech-
nique similar to that developed by Marchenko et al. (2015) for $NO_2$ SCD retrieval. The fitting proce-
dure derives the $O_2$-$O_2$ SCD using retrieved $O_3$ and $NO_2$ slant column estimates from independent
OMI algorithms.

The OCP is estimated using the MLER method to compute the appropriate air mass factors (AMF)





(Yang et al., 2015). To solve for OCP, we invert

$$\text{SCD} = \text{AMF}_g(P_s, R_g)\text{VCD}(P_s)(1 - f_r) + \text{AMF}_c(\text{OCP}, R_c)\text{VCD}(\text{OCP})f_r, \tag{5}$$

where VCD is the vertical column density (VCD=SCD/AMF), $\text{AMF}_g$ and $\text{AMF}_c$ are the precom-
puted (at 477 nm) clear sky (ground) and overcast (cloudy) subpixel AMFs, $P_s$ is the surface pres-
sure, and $f_r$ is the cloud radiance fraction (CRF) given by $f_r = \text{ECF} * I_c/I_m$. Lookup tables of
the TOA radiances and AMFs were generated using VLIDORT. Temperature profiles needed for
computation of VCD and AMF are taken from the Global Modeling Initiative (GMI) chemistry
transport model (Strahan et al., 2007) driven by the NASA GEOS-5 global data assimilation system
(Rienecker et al., 2011). Comparisons of the retrieved OCPs with those from the operational KNMI
OMI $O_2$-$O_2$ algorithm, OMCLDO2, have shown good agreement with a correlation coefficient of
$\sim$0.99 for ECF$> 0.2$ when identical surface climatological LERs are used.

### 2.3.2 OMI $NO_2$ algorithm

The OMI $NO_2$ spectral fitting algorithm (OMNO2A) currently uses differential optical absorption
spectroscopy (DOAS) to fit OMI-measured spectra in the wavelength range 405–465 nm to estimate
total (stratospheric and tropospheric) $NO_2$ SCDs (Boersma et al., 2011). The SCDs are then con-
verted to $NO_2$ VCDs using pre-calculated AMFs: VCD=SCD/AMF using the algorithm known as
OMNO2B (Bucsela et al., 2013; Lamsal et al., 2014). For fixed (measured) SCD, the retrieved $NO_2$
VCD is inversely proportional to the AMF.

## 3 Basic approach

The BRDF over land is calculated as $\text{BRDF} = a_{iso} + a_{vol}k_{vol} + a_{geo}k_{geo}$, where the coefficients
$a_{iso}, a_{vol}, a_{geo}$ come from MODIS data, the volumetric kernel, $k_{vol}$ describes light reflection from
a dense leaf canopy, and the geometric kernel, $k_{geo}$ describes light reflection from a sparse ensemble
of surface objects casting shadows on the background assumed to be Lambertian. The kernels are
the only angle-dependent functions, which expressions are given in Lucht et a. (2000). The BRDF
coefficients are spatially averaged over an actual satellite FOV and used to calculate TOA radiance
for its observation geometry.

The BRDF coefficients depend on wavelength. For the present study we selected two wavelengths
in the UV and Vis: 354 and 466 nm. These wavelengths are relatively free of atmospheric rotational
Raman scattering (RRS) and trace gas absorption. The BRDF coefficients at 466 nm are directly
taken from the MCD43GF product at 470 nm that is provided at a spatial resolution of 30 arc sec
(Schaaf et al., 2002, 2011). Because the MODIS product is not available at 354 nm, we adjusted
the 470 nm LERs to account for potential spectral dependences. The adjustment applies the spectral
ratio of climatological OMI-derived LERs: $R(354)/R(470)$ similar to the approach of McLinden et



al. (2014). Using climatological data of Kleipool et al. (2008) we find that this ratio is close to unity (within $\pm 5\%$) for most areas.

To calculate TOA radiance over water surfaces, we account for both light specularly reflected from a rough water surface and diffuse light backscattered by water bulk and transmitted through the water surface. We neglect contributions from oceanic foam. Reflection from the water surface is described

by the Cox-Munk slope distribution function (Cox and Munk, 1954). We use an isotropic form of the Cox-Munk distribution in which the facet-slope variance is independent of wind direction. All computations use a wind speed of 5 m/s which is close to the climatological mean.

Diffuse light from the ocean is described by a Case 1 water model that has chlorophyll concentration as a single input parameter (Morel, 1988). Our Case 1 water model accounts for the anisotropic

nature of light backscattered by the ocean (Morel and Gentili, 1996). A spatial distribution of chlorophyll concentration is taken from the monthly SeaWiFS climatology. The common Case 1 water model developed for the Vis (Morel, 1988) was extended to the UV using data from Vasilkov et al. (2002, 2005). To calculate water-leaving radiance, we need to know the downwelling atmospheric transmittance at the surface. The transmittance is obtained by calculating the total atmospheric direct

and diffuse downwelling flux at the surface. The diffuse contribution in the transmittance will itself depend on the water-leaving radiance. To calculate the atmospheric transmittance, we introduce in VLIDORT a module for the iterative calculation of the transmittance, in which the first computation is made for a dark surface, and this is then used again as input to the water-leaving contribution. This process is repeated until convergence of the transmittance is achieved (3 or 4 iterations are

sufficient).

To estimate LER over over mixed surface types, we compute an area-weighted radiance for uniform land and water contributions within an OMI FOV. The LER for heterogeneous surface pixels is then calculated from this linear combination of radiances. The high spatial resolution MYD43GF product supplies an eight category land water classification map at the same resolution as the BRDF

parameters. We convert this map into a binary land-water mask by merging all shorelines and ephemeral water into the land category and classifying all other water sub-categories simply as water. We then compute the areal fraction of land and water for each OMI FOV. For specification of the OMI pixel, we used the OMPIXCOR product that provides coordinates of OMI pixel corners (http://disc.sci.gsfc.nasa.gov/uui/datasets/OMPIXCOR_V003/summary). We used an option of

overlapping pixels in the along track direction corresponding to 75% energy in the along-track FOV. In this option the edges of the FOV are aligned in the cross track direction but overlap in the along track direction.

Given the computed TOA radiance, $I_{TOA}$, the LER is calculated by inverting

$$I_{TOA}(\lambda, \theta, \theta_0, \phi, P_s, R) = I_0(\lambda, \theta, \theta_0, \phi, P_s) + \frac{RT(\lambda, \theta, \theta_0, P_s)}{1 - RS_b(\lambda, P_s)}, \qquad (6)$$

where $\lambda$ is wavelength, $\theta$ is the VZA, $\theta_0$ is the SZA, $\phi$ is the relative azimuth angle, $R$ is the LER, $I_0$ is the TOA radiance calculated for a black surface, $T$ is the quantity representing the total trans-



mittance in the viewing direction when the atmosphere is illuminated from below by a Lambertian reflection of all energy incident upon the ground, and $S_b$ is the diffuse flux reflectivity of the atmosphere for the case of its isotropic illumination from below (Chandrasekhar, 1960; Dave, 1978).

To speed up computations, we created lookup tables of the quantities $I_0$, $T$, and $S_b$ for selected wavelengths.

## 4   Geometry-dependent LER

Because reflection of incoming solar light from non-Lambertian surfaces depends on satellite observational geometry, the same area observed at different geometries can have different LERs. Figure 1

shows the MODIS-based high spatial resolution LER over the US Baltimore-Washington corridor for two consecutive days (Jan. 17 and 18, 2005) computed using the OMI observational geometry. The SZA and VZA values are in the similar ranges for both days. However, there is a large difference in the relative azimuth angle which varies from around 63° for Jan. 17 to about 118° for Jan. 18. Since the land tends to have strong backward scattering, that explains the higher LER for Jan.

18 than that for Jan. 17. The differences, if not accounted for, may produce errors in the trace gas retrievals.

A comparison of the computed geometry-dependent and climatological LERs at 466 nm is shown in Fig. 2 for OMI orbit 12414 on 13 Nov 2006. The climatological LERs (monthly) are derived from OMI observations (Kleipool et al., 2008). In general, the eastern portion of the orbital swath (that

has a later equator crossing time) has higher values of the LERs than the western part. This is an effect of the OMI observational geometry and BRDF increase in the backscattered direction.

Figure 2 shows significant differences between the geometry-dependent and climatological LERs for both land and ocean. Over land, the climatological LERs are mostly higher than the geometry-dependent LERs. This is presumably because the geometry-dependent LERs are derived from at-

mospherically corrected MODIS radiances while the climatological LERs are affected by residual aerosols. Moreover, climatological LERs are inherently contaminated by clouds owing to substantially larger sizes of OMI pixels as compared with those of MODIS. This is particularly true for the Amazonia region where clouds are persistent.

Over ocean, the geometry-dependent LERs are systematically higher than the climatological LERs

in areas affected by sun glint and at large VZAs. This is because the climatological LERs are based on the mode of LERs from a long time series of observations over a given area; this minimizes the impact of observations affected by sun glint and high values that occur at large VZAs.

Figure 3 shows the geometry-dependent LERs computed at 466 and 354 nm and their differences for same OMI orbit 12414. Here, we assume that the BRDF coefficients over land are spectrally

independent. The LER differences over land are thus solely due to the smoothing effect of enhanced Rayleigh scattering in UV that increases the diffuse to direct incident irradiance ratio as compared



with 466 nm. Over land, LER(354) < LER(466), but the differences are relatively small (< 0.015).

Over the ocean, the LER differences additionally result from the spectral dependence of water-leaving radiance. Over the sunglint areas, the solar light reflected from the ocean surface is significantly brighter at 466 nm than at 354 nm thus leading to higher LERs. Over areas less affected by sunglint, LER(354) > LER(466) in general owing to higher amounts of water leaving radiance.

It is interesting to note that the patterns of rivers and their tributaries are evident in the LER maps of Fig. 3 for both 354 and 466 nm. This effect is most pronounced when rivers are viewed from the OMI measurement geometry that registers the reflectance signal of Fresnel reflection from smooth river surfaces. It may be somewhat surprising that this appears at OMI spatial resolution; we can explain the effect by considering that while the LER from FOVs comprised of river areas and surrounding land is weighted linearly by the areal fraction of each, reflectance from the river surface is disproportionally high owing to the Fresnel reflection in sun-glint geometry. Outside of the regions where OMI observes glint, the LER in the Amazon basin may still be higher than expected owing to the turbidity of some rivers in the Amazon floodplain that varies seasonally.

## 5 BRDF effects on the OMI cloud products

### 5.1 RRS algorithm

Figure 4 shows ECFs computed with geometry-dependent LERs and the differences with respect to the climatological LERs ($\Delta$ECF). The largest $\Delta$ECFs (up to 0.05) take place over the less cloudy Amazonian areas. $\Delta$ECF is obviously lower for cloudy areas owing to the diminished effect of surface properties on TOA radiance. The heavily cloudy areas are easily identified on the $\Delta$ECF map.

We next examine the most interesting range of ECF for trace-gas retrievals, ECF < 0.25, which corresponds to $f_r <$ 0.4–0.5. For this range, Figure 5 shows a scatter plot of the ECFs retrieved with the geometry-dependent versus climatological LERs and how $\Delta$ECF varies with ECF. Only data from 50°S to 50°N are used in Fig. 5 and all subsequent similar figures. This latitude range excludes areas with snow for which MODIS BRDF data are not available. On average, $\Delta$ECF is small and positive for the ocean ($\sim$ 0.02). Over land $\Delta$ECF is even lower and ranges from $\sim$-0.01 to $\sim$0.015 for ECF< 0.25. The standard deviation of $\Delta$ECF does not depend much on ECF. It is $\sim$0.01 over ocean and $\sim$0.015 over land. Even though $\Delta$ECF is small on average, it can be as large as $\pm$0.05 which is quite substantial for the low ECF range.

Figure 6 similarly shows OCPs retrieved with the geometry-dependent LER and the differences with respect to those retrieved using the climatological LERs ($\Delta$OCP) for OMI orbit 12414. There are no obvious geographical patterns in the $\Delta$OCP map. $\Delta$OCP can be as large as $\pm$100 hPa. The OCP differences are particularly pronounced along the edges of cloud systems. Spatial correlation between $\Delta$OCP (Fig. 6) and $\Delta$ECF (Fig. 4) is not apparent. As may be expected, $\Delta$OCP decreases



with increasing ECF. Figure 7 is similar to Fig. 5 but for OCP. On average, $\Delta$OCP is small ($\sim$ 10.0 hPa) with standard deviation of up to $\sim$40 hPa.

### 5.2  $O_2$-$O_2$ algorithm

Here we show similar comparisons of the cloud products retrieved with the geometry-dependent and climatological LERs for ECF< 0.25. Figure 8 is similar to Fig. 5 but for ECF from the $O_2-O_2$ algorithm. $\Delta$ECF $<\sim$0.03 over land and $<\sim$0.01 over ocean.

Figure 9 is similar to Fig. 7 but for OCP from the $O_2-O_2$ algorithm. $\Delta$OCP has values up to 200 hPa. The mean $\Delta$OCPs are significantly larger for the $O_2$-$O_2$ algorithm as compared with RRS.

On average, $\Delta$OCP varies from $\sim$80 hPa at ECF=0.05 to 5 hPa at ECF = 0.25 over land. $\Delta$OCP is noticeably lower over ocean. The standard deviation, up to 100 hPa, is also higher than that from the RRS cloud algorithm. This can be explained by decreasing Rayleigh optical thickness at 477 nm, which results in a larger fraction of direct solar irradiance illuminating the surface and larger BRDF effects.

### 6  BRDF effects on the OMI $NO_2$ retrievals


We consider the BRDF effect on the $NO_2$ AMFs only, because the retrieved $NO_2$ amount is inversely proportional to the AMF. The tropospheric $NO_2$ AMF, $AMF_{\text{trop}}$, is calculated using the MLER model with input cloud parameters from the $O_2-O_2$ algorithm assuming *a priori* $NO_2$ vertical profile shapes (see Fig. 10):

$$AMF_{\text{trop}} = AMF_g(P_s, R_g)(1 - f_r) + AMF_c(\text{OCP}, R_c)f_r. \tag{7}$$

The effect of a surface reflectivity change, $\Delta R_g$, of 0.01 on $AMF_g$ is shown as a function of $R_g$ in Fig. 10. The Jacobian, $J = \Delta AMF_g/\Delta R_g$, is always positive because larger surface reflectances increase satellite sensitivity to $NO_2$ absorption in the lowest atmosphere. $J$ decreases with increasing $R_g$ and for unpolluted $NO_2$ mixing ratios (Fig. 10).

The BRDF effect on $AMF_g$ for OMI observational geometries and ground resolution can be estimated from Figures 2 and 10 using $\Delta R_g = \text{LER(BRDF)} - \text{LER}$. The effect is largest over polluted regions in the eastern US, where $\Delta R_g$ is negative with values –0.03 to –0.02 (Fig. 2), LER $\sim$ 0.05 and $\Delta AMF_g \sim$ –20% to –30%. The BRDF effect reverses over water for glint geometries and large viewing angles, but $R_g$ is large here and the effect on $AMF_g$ is reduced (i.e., small Jacobian).

To estimate the BRDF effect on $AMF_{\text{trop}}$ we need to account for the $f_r$ change as well:

$$\Delta AMF_{\text{trop}} = \Delta AMF_g(R_g)(1 - f_r) + \Delta f_r[AMF_c(\text{OCP}, R_c) - AMF_g(R_g)]. \tag{8}$$

The cloud AMF strongly depends on the OCP, since high clouds (low OCP) have a shielding effect and low clouds (high OCP), aerosols, and fog can enhance $AMF_c$. Assuming a negligible $NO_2$



mixing ratio above the cloud OCP, we can neglect $AMF_c$ and Eq. 8 simplifies to

$$\Delta AMF_{trop} = \Delta AMF_g(R_g)(1 - f_r) - \Delta f_r AMF_g(R_g) \qquad (9)$$

Over land, BRDF reduces the geometry-dependent LER as compared with the LER climatology, (i.e., $\Delta R_g < 0$ ) leading to smaller values of $AMF_g$ (Fig. 10). At the same time, the mean ECF increases by 0.02 (Fig. 8) and this produces even larger increases in $f_r$ ($\Delta f_r \sim 0.04$). Therefore both terms in the above equation are negative meaning that switching to a geometry-dependent LER reduces $AMF_{trop}$ even more over land. The effect is mixed over water, since both $\Delta R_g$ or $\Delta f_r$ can change signs for certain geometries.

Figure 11 shows that the calculated BRDF impact on $AMF_{trop}$ arising from both surface BRDF and $O_2$-$O_2$ cloud parameters exhibits a strong spatial variation with smaller effects over ocean, unpolluted, or cloudy areas. Over land, where the geometry-dependent LER is generally lower than the climatological LER, use of the BRDF data results in lower AMFs and higher tropospheric $NO_2$ VCDs. The effect is enhanced over polluted areas such as eastern US, where the changes in AMF can reach up to 50%. The effect is reduced for unpolluted and overcast conditions and mixed over oceans, because $R_g$ increases for sunglint and large VZA directions but decreases for other directions.

Figure 12 compares the clear-sky $AMF_{trop}$ calculated using climatological and geometry-dependent LERs. The use of geometry-dependent LERs generally leads to lower $AMF_{trop}$ by up to 29% over land and 15% over ocean. Differences in $O_2$-$O_2$ cloud parameters resulting from the use of geometry-dependent LERs add additional scatter, changing $AMF_{trop}$ by -42–5% over land and -22–13% over ocean. $AMF_{trop}$ differences are large for low AMFs, driven by enhanced differences in LER, OCP, or $f_r$.

Figure 13 illustrates how the use of geometry-dependent LER changes $NO_2$ retrievals over clean and polluted areas. Consistent with previous studies by Lin et al. (2014, 2015), AMFs are considerably lower with geometry-dependent LERs. This suggests that the current operational $NO_2$ products based on climatological LERs could be underestimated by up to 48% over China. The eastern US exhibits similar but somewhat smaller differences. Minor changes are expected over unpolluted and overcast areas.

## 7 Conclusions

We developed a new concept of geometry-dependent surface LER and provided a means for computing it. Spatially averaged high-resolution MODIS BRDFs are used for computation of the geometry-dependent LER over land for OMI pixels. The Cox-Munk slope distribution function and the Case 1 water-leaving radiance model are utilized for computation of the geometry dependent LER over ocean. This method accounts for the geometrical dependence of LER within the existing framework of MLER trace gas and cloud algorithms with only minimal changes. It is important to note





that the geometry-dependent surface LER approach can be applied to any current or future satellite
algorithms that utilize MLER trace gas and cloud algorithms.

We examined the effects of the geometry-dependent LER on OMI cloud and $NO_2$ algorithms. The
effects on retrieved cloud parameters were relatively small on average and diminish with increasing
cloud fraction. Even though the impact is small on average, it can be as large as $\pm 0.05$ for the
effective cloud fraction and $100\,hPa$ for the cloud optical centroid pressure. The BRDF effects were
noticeably higher for the $O_2$-$O_2$ algorithm that uses visible wavelengths as compared with the RRS
algorithm that utilizes a UV spectral range. This can be explained by the stronger smoothing effect
of Rayleigh scattering in the UV as compared with the Vis.

We also find that the use of geometry-dependent LER increases the OMI $NO_2$ vertical columns by
up to 50% over highly polluted areas. Only minor changes to $NO_2$ columns (within 5%) are found
over unpolluted and overcast areas.

In the future, we plan to implement the use of geometry-dependent LERs in our cloud and $NO_2$
OMI algorithms. Along with the use of the geometry-dependent LER product, we plan to explicitly
include aerosols in the $NO_2$ algorithm. Further evaluation of the results with OMI data is ongoing.
We also plan to investigate the use a new surface BRDF product from the multi-angle implementation
of atmospheric correction (MAIAC) algorithm (Lyapustin et al., 2012).

*Acknowledgements.* Funding for this work was provided in part by the NASA through the Aura science team
program. We thank P. K. Bhartia for helpful discussions, Z. Ahmad for providing data for comparisons, A. Sayer
for provision of an updated ocean optics model used in the water-leaving supplement of the VLIDORT code,
and C. B. Schaaf for consultation on the use of the MODIS BRDF product.



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





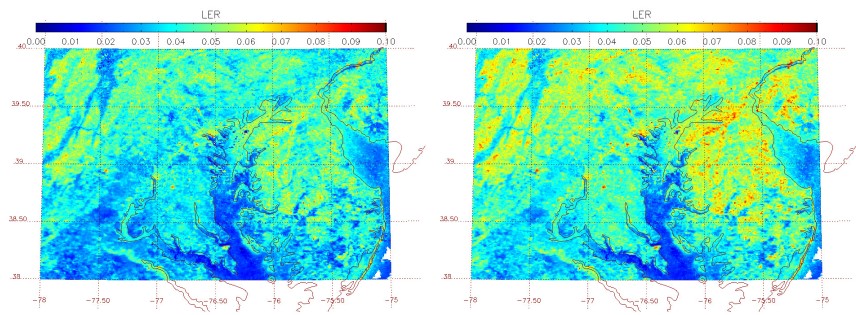

**Fig. 1.** High spatial resolution MODIS-based LERs for the Baltimore-Washington corridor for 17 (left) and 18 (right) Jan. 2005 computed for OMI observational geometries.





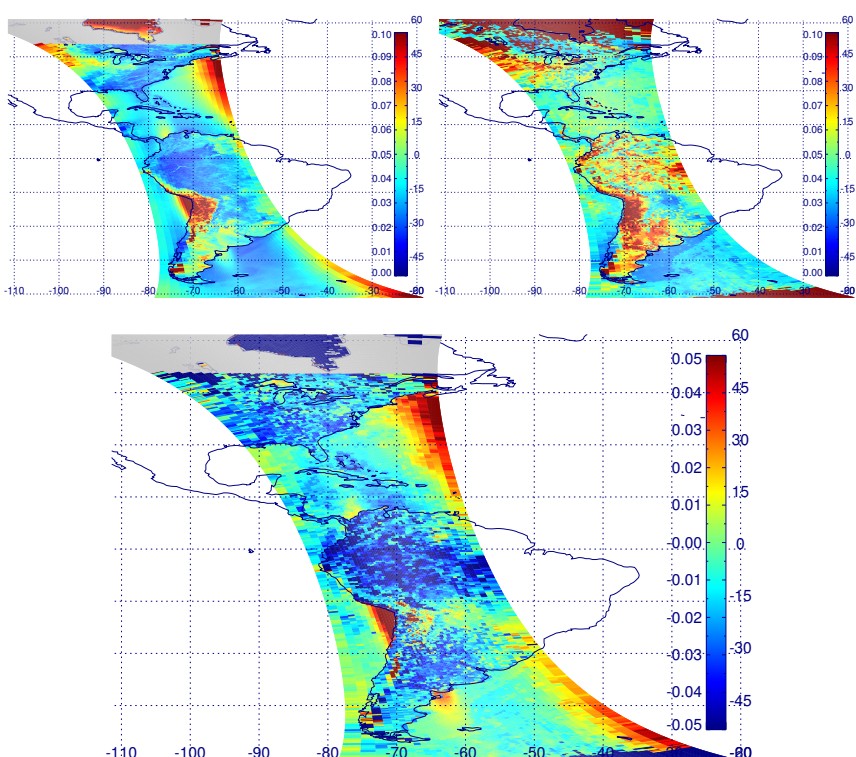

**Fig. 2.** LERs computed at 466 nm for OMI orbit 12414 on 13 Nov. 2006 using MODIS-based BRDF with OMI geometry (upper left), OMI-based monthly climatology (upper right), and their difference (MODIS-based minus climatological LERs, lower panel) Missing MODIS BRDF data are shown in grey here and elsewhere.





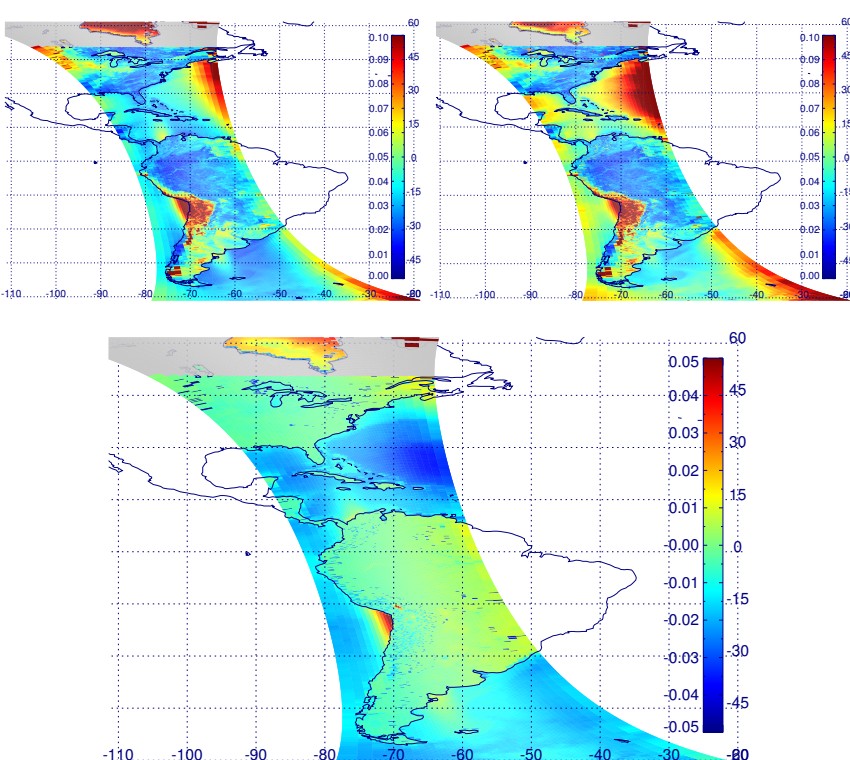

**Fig. 3.** Similar to Fig. 2 but showing geometry-dependent LERs computed for 466 nm (upper left), 354 nm (upper right), and their difference (466 nm minus 354 nm LER, lower panel).





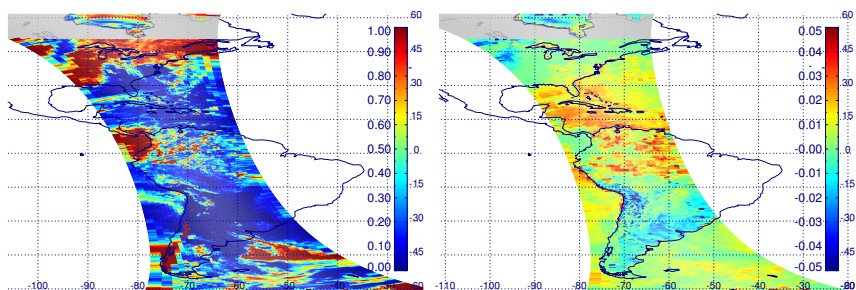

**Fig. 4.** RRS-derived ECF computed with geometry-dependent LERs (left) and the difference between the ECFs computed with geometry-dependent and climatological LERs (right).





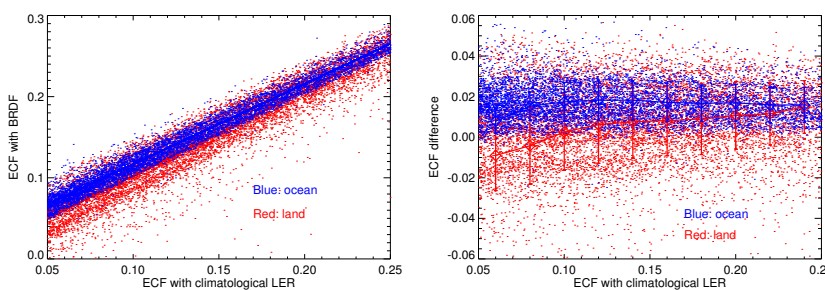

**Fig. 5.** Scatter plot of RRS-retrieved effective cloud fractions (ECFs) computed with geometry-dependent LERs versus climatological LERs for ECF< 0.25 with linear fits (left), and the mean ECF difference (diamonds) and standard deviation (error bars) as a function of ECF (right).





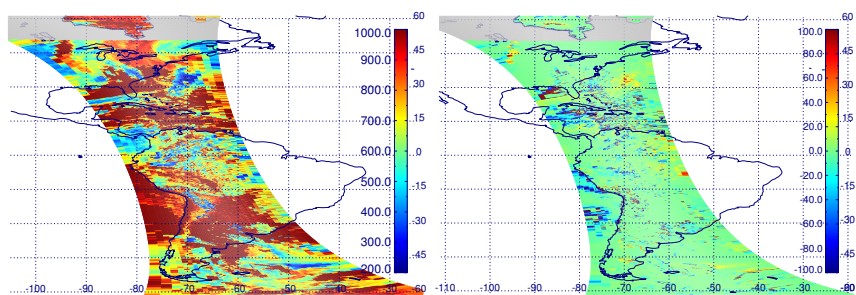

**Fig. 6.** RRS-retrieved cloud optical centroid pressure (OCP) computed with geometry-dependent LERs (left) and the difference between the OCPs computed with geometry-dependent and climatological LERs (right).





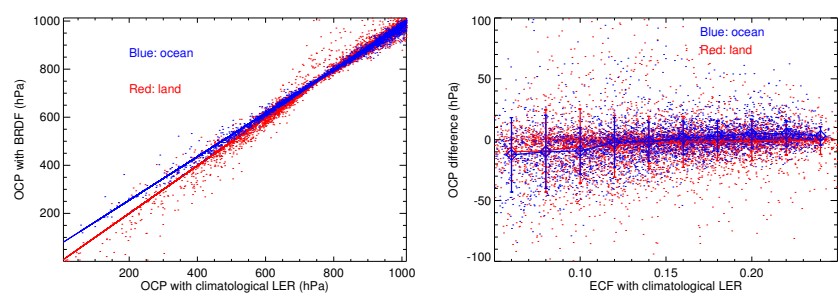

**Fig. 7.** Similar to Fig. 5 but for cloud optical centroid pressures (OCP).





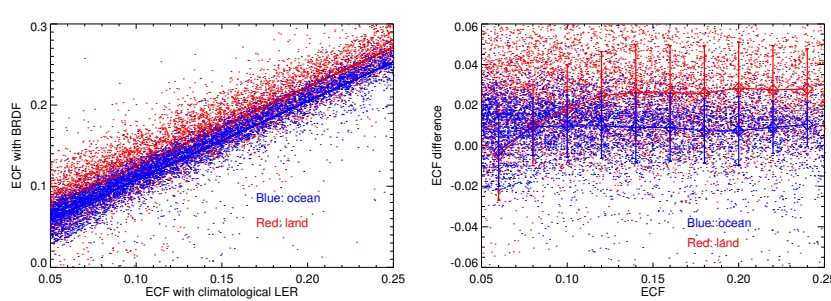

**Fig. 8.** Similar to Fig. 5 but for effective cloud fraction (ECF) from the $O_2$-$O_2$ algorithm.





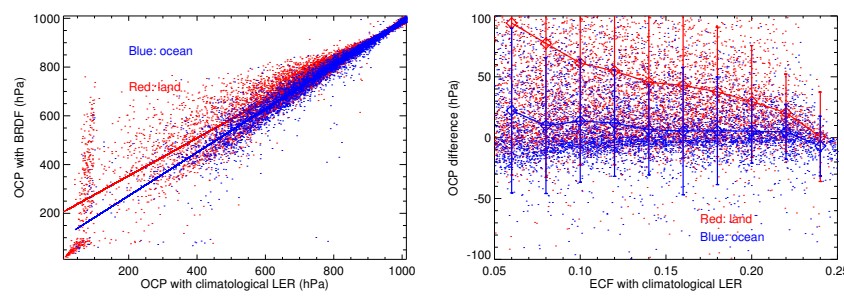

**Fig. 9.** Similar to Fig. 7 but for cloud optical centroid pressure (OCP) from the $O_2$-$O_2$ algorithm.





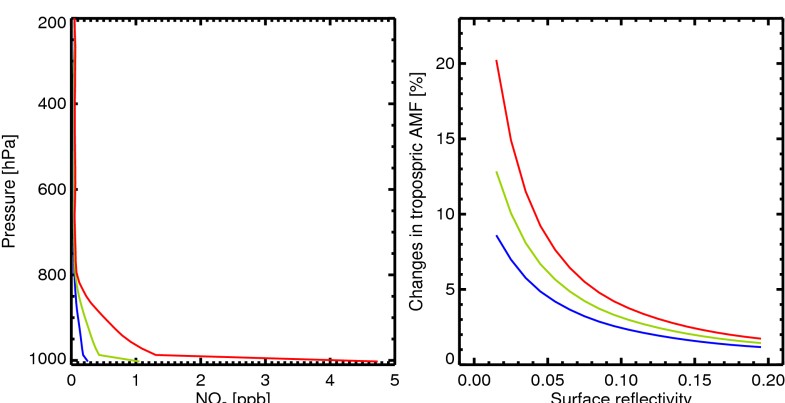

**Fig. 10.** June mean $NO_2$ profiles at three locations in the eastern US from the NASA GMI model (left) and air mass factor (AMF) change due to 0.01 change in reflectivity as a function of surface reflectivity (right); Red: highly polluted profile, green: moderately polluted, blue: unpolluted profile.





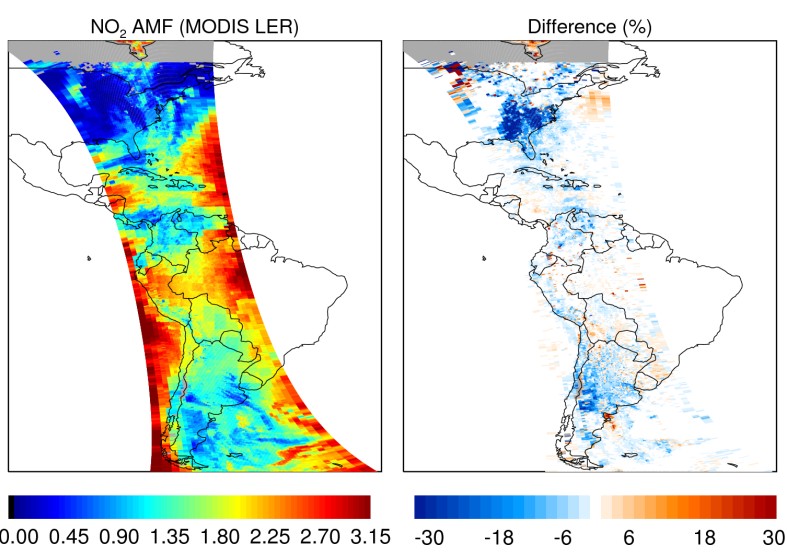

**Fig. 11.** OMI tropospheric $NO_2$ air mass factor (AMF) calculated using geometry-dependent MODIS-based LER (left) and percent differences with respect to climatological LERs (right).





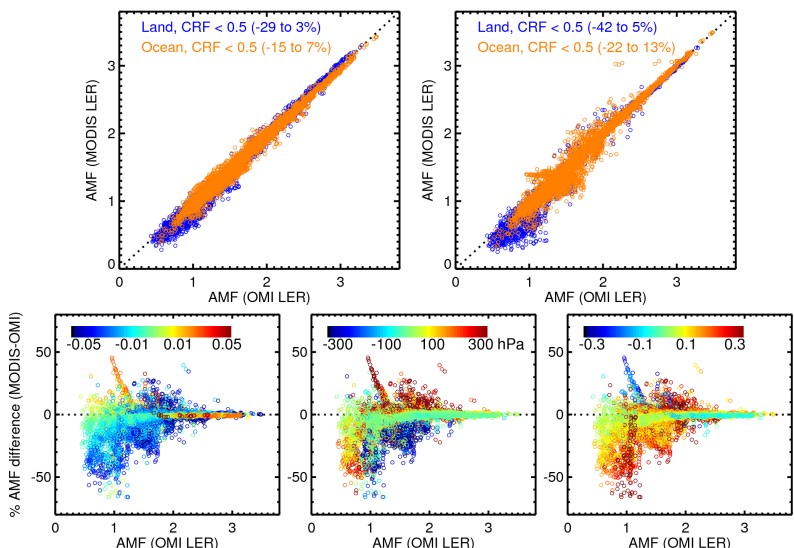

**Fig. 12.** Top panels: Scatter diagrams of AMFs calculated using geometric-dependent MODIS-based LER versus OMI-based climatological LER for the orbit 12414 for clear to moderately cloudy sky ($f_r < 0.5$) including effects of BRDF only (clouds unchanged, left) and the effects of both BRDF and $O_2$-$O_2$ cloud parameters (right) for land (blue) and ocean (orange). Numbers in parentheses represent % difference at the 2nd and 98th percentile range. Bottom panels: % difference in AMF with changes in surface BRDF and $O_2$-$O_2$ cloud parameters, sorting the data by the difference with respect to LER (left), OCP (middle), and $f_r$ (right).



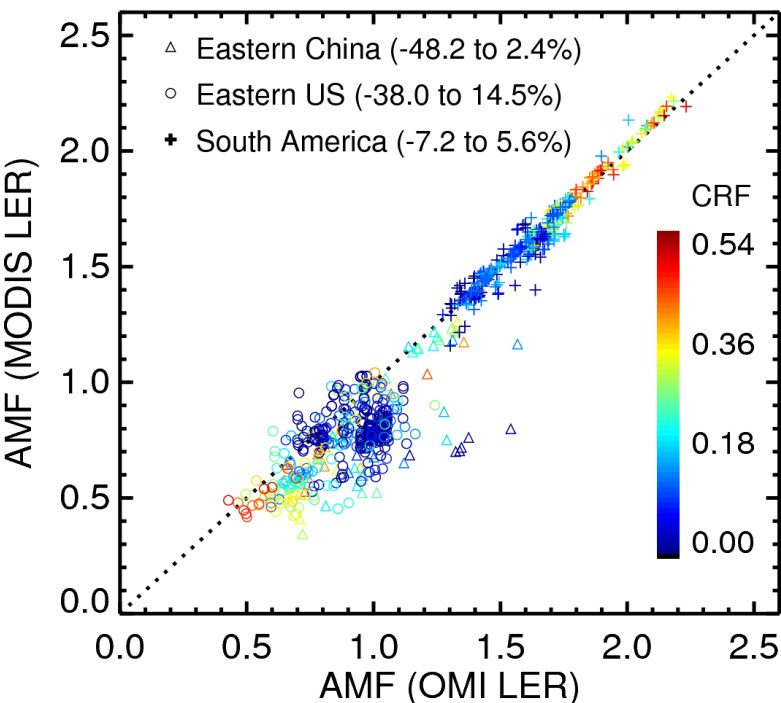

**Fig. 13.** AMFs calculated with geometry-dependent MODIS-based LERs and climatological OMI-based LERs over 5°x5° boxes in eastern China (115°–120°E, 36°–41°N, triangle), eastern US (75°–80°W, 36°–41°N, circle), and South America (55°–60°W, 20°–25°S, plus sign) for clear to moderately-cloud skies $f_r < 0.5$. AMF calculated with the MODIS-based LER includes the combined effects of surface BRDF and $O_2$-$O_2$ cloud parameters. Symbols are color-coded by $f_r$. Numbers in parentheses represent % differences at the 2nd and 98th percentile ranges.