# Peer review of "Accounting for the effects of surface BRDF on satellite cloud and trace-gas retrievals: A new approach based on geometry-dependent Lambertian-equivalent reflectivity applied to OMI algorithms"

_Atmospheric Measurement Techniques, 2016_

## Referee Comment (RC1) · Anonymous Referee #1 · 30 May 2016

General

The topic of this paper is important for satellite retrieval of NO2 and other air pollutant gases. To precisely calculate the observed light path of trace gases, the inclusion of the bidirectional surface reflection has to be taken into account. Here it is proposed to sample the BRDF for the actual sun-satellite geometry to get a better estimate for the LER. The approach looks promising, and the results are interesting. After addressing the points below, the paper may be accepted.

[Figure]

The paper is well readable. However, at some points clarification is needed. A concern is the start of the introduction, which may lead to confusion. Furthermore, the interpretation of the results needs improvement. This is all indicated below.

Main points

(1) The start of the introduction, from l. 26 – l. 52 around Eqs. 1-3, seems detached from the remainder of the paper. The symbols and terms are different. The text and formulae are unclear. What is the link to the LER, which is used in the remainder of the paper?

- What is $I(\omega)$ in Eq. 1? Is that the same as $I_m$ in Eq. 4? What is the relation to the top-of-atmosphere radiance as observed by OMI ? Why do you use solid angle $\omega$, whereas in the remainder of the paper you use $\theta$ and $\phi$? Why is F, mentioned below Eq. 1, not used in the equation? Explain $\theta_r$, which is called $\theta$ in the remainder of the paper.

- In Eq. 2 please give the integration limits. Below Eq. 2 it is apparently assumed that this particular Lambertian has an albedo of 1. But also for a less reflective Lambertian surface the relation can be used.

- Eq. 3: does BRF in this equation yields Rg in Eq. 4?

(2) The interpretation of the scatter plots of retrieved cloud parameters from RRS and O2-O2 algorithms between the BRDF-dependent LER and climatology LER, especially the OCP, deserves more discussion. See Figs. 7 and 9. Apparently the RRS OCP is hardly depending on the surface BRDF, whereas the O2-O2 OCP is strongly depending on it. That is remarkable. It cannot be only explained by the decrease of Rayleigh scattering at 466 nm as compared to 354 nm, as stated in the paper. Another difference in both algorithms must be causing this. It is probably due to the fact that the RRS signal is not including all light paths that are relevant for the O2-O2 absorption (and NO2 absorption). Namely, the direct light path of direct sunlight reflected by the surface

and arriving at the satellite is not included in the RRS signal, because there is no Rayleigh (Raman) scattering involved. But it is an important light path for the cloud-free part of the pixel. And this direct light path is also strongly contributing to O2-O2 (and NO2) absorption. So the RRS method is in first order insensitive to the surface and to its BRDF. Only via the light paths Rayleigh + surface reflection, and surface reflection + Rayleigh can the RRS signal pick up surface BRDF effects. But that is a second-order effect. Please consider this cause in explaining the OCP behaviour of the RRS and O2-O2 algorithms.

(3) Please add histograms of ECF and OCP for the orbits shown, and not only scatter plots, to see the difference between including and excluding BRDF effects, and the difference between RRS and O2-O2 algorithms.

Questions and textual comments

- Eq. 4: please say that I_g and I_c are at top-of-atmosphere

- L. 67: is Ac=0.8 also assumed in this paper?

- L. 71: add here a reference to Stammes et al. (2008)

- L. 76: add here a reference to Sneep et al. (2008)

- L. 121: remove: its

- L. 141: does the RRS ECF hold for Rc=0.8 ?

- L. 175: please make a separate equation of the in-text formula.

- L. 185: please clarify: do you use in the paper the climatological ratio Rg(354)/Rg(470) or a ratio of unity?

- L. 208: please give a reference for MYD43GF.

- L. 215: I_TOA: why is a new symbol introduced? Where the other radiance symbols not at top-of-atmosphere? How does it relate to I_m of Eq. 4? Please do not introduce unnecessarily new quantities and symbols. Please also relate \theta, \phi, and \theta_0 to the earlier introduced angles.

- L. 217 ff: the explanation of T is unclear. T is the total two-way transmission of the atmosphere.

- L. 245: Land is mostly darker in the UV than in the VIS. So why not use the climatological OMI data base at 354 nm?

- L. 265: please indicate the orbit and date.

- L. 300: is in Sect. 6 only the O2-O2 algorithm used and not the RRS algorithm because the latter has very little impact of BRDF?

- L. 301: why are the NO2 profile shapes from June and not from November, for which month the satellite data were chosen?

- L. 313: can you please explain how this formula is derived?

- L. 355: please mention here that the background aerosols are included in the climatological LER, but are missing in the BRDF, so that the ECF from the BRDF has a low bias.

- L. 366: the use > to use

- L. 385: missing: Chandrasekhar

Figures and captions:

Fig. 1: please use larger font for lat/lon (like in Fig. 2). What is the spatial resolution of these maps?

Fig. 2: please use a, b, c for the subplots. This also holds for the other figures with 3 subplots.

Fig. 4: which orbit and date? With which LER figure should this be compared? RRS-derived > RRS-retrieved

FIg. 5: which orbit and date?

Fig. 7: please write out the caption.

For Fig. 7 and Fig. 9 please consider inverting the axes, thus from 1000 to 0 hPa, because that looks more natural (low clouds at the origin of the plot).

Fig. 9: Please use better caption; the reference to the caption of Fig. 5 leads to another reference to another caption. What are the straight lines in the left plot?

Fig. 10: reflectivity > surface reflectivity

Fig. 11: which orbit and date?

Fig. 12: please number the subplots. Please add LER, OCP, fr to the legend of the lower 3 subplots.

Fig. 13: which date and orbit?

─────────────────────────────

---

## Referee Comment (RC2) · Anonymous Referee #2 · 3 Jun 2016

Surface reflectance is an important parameter in satellite trace gas retrievals in the UV/VIS range. This study proposes an approach that is able to account for surface BRDF effects while requiring only minimal changes to existing retrieval algorithms.

This is a very welcome contribution to this topic. The publication is generally well written and clear and the methods are sound. Particularly novel and valuable aspects of the manuscript are an approach for considering ocean BRDF in cloud and trace gas retrievals and a comprehensive analysis of BRDF effects on cloud parameters.

Despite these positive aspects, the study has serious deficiencies as detailed below that need to be addressed before it can be considered for publication. This will require a major revision.

Major issues:

There have been several previous studies (e.g. Russel et al., 2011; McLinden et al., 2014; Kuhlmann et al., 2015) comparing NO2 columns retrieved with MODIS-based surface reflectance/albedo with NO2 columns retrieved using climatological Lambertian Equivalent Reflectances (LER) such as the Kleipool et al. (2008) data set.

As properly referenced, there have also been several previous studies on the effect of surface reflectance anisotropy (BRDF effects) on NO2 retrievals (Zhou et al., 2010; Noguchi et al., 2014; Lin et al., 2014, 2015).

The novelty of this study as compared to these earlier ones is the treatment of the surface reflectance as a geometry-dependent LER to account for BRDF effects. To be a valuable contribution to the existing body of literature, the publication needs to demonstrate the advantages and limitations of this approach, but it falls short in doing so for two reasons:

1) Instead of analyzing the effects of geometry-dependent versus geometry-independent LER it only compares results based on MODIS reflectance products with results based on OMI-LER from Kleipool et al. (2008). These results are only little influenced by the geometry-dependence of surface reflectance but are dominated by the large differences between MODIS and OMI-based reflectance data sets, an aspect that has been addressed at length in previous studies. For the same reason, the conclusion in the abstract (and conclusions section) that geometry-dependent LERs can change NO2 vertical columns by up to 50% is very misleading as it gives the wrong impression that this is a direct consequence of the geometry-dependence.

In contrast to the present study, Zhou et al. (2010) differentiated between the effects of switching from an OMI climatological LER to a MODIS reflectance product from the effects of considering BRDF effects versus not considering such effects. They concluded that considering geometry-dependent versus geometry-independent reflectance changes NO2 by mostly well below 20% and that these changes are smaller than those induced by switching from the OMI-based LER of Kleipool et al. (2008) to MODIS.

2) The proposed approach of using a geometry-dependent LER instead of a full BRDF treatment is very interesting since it has the potential to simplify the retrieval (e.g. existing look-up-table based retrievals could still be used) while accounting for the influence of surface anisotropic reflectance at least to first order.

However, this is still a simplification with respect to a full BRDF treatment since only the TOA radiance is reproduced but the paths of photons reaching the TOA are not exactly the same as in the case of a full BRDF treatment with likely consequences on the vertical sensitivity profile (box AMF profile). The study fails to demonstrate the implications of such a simplification. The effects of such simplifications were addressed by Zhou et al. (2010) which compared a full BRDF treatment with a treatment taking either the MODIS albedo as LER or taking the BRF value for the given illumination and viewing geometry as LER. In both cases, differences from a full BRDF treatment were significant (see their Fig. 10). A similar analysis is needed for the approach proposed here in order to demonstrate both the advantages of a geometry-dependent LER as compared to a geometry-independent LER (Fig. 1 provides some hints) and the limitations with respect to a full BRDF treatment.

Minor points:

Page 2, Line 24: I suggest to include the MERIS based albedo data set of Popp et al. (2011) which is used in the latest FRESCO cloud algorithm and will be considered also for future TROPOMI products.

P5, L152-155: Please explain why O3 and NO2 slant columns are taken from independent OMI algorithms.

P6, L159: It would probably be useful to make clear that you are referring to vertical column densities of O2-O2.

P7, L211: "over over" -> "over"

P7, Equation (6): This equation appears incomplete as the rightmost term only multiplies two unit-less numbers (reflectance times transmittance) but does not represent a radiance.

P8ff: The manuscript structure would probably become clearer by introducing a new section "4 Results and Discussion" and making the present sections 4-6 subsections of this.

P8, L244-245: This is only true over land, not over the ocean.

P8, L254ff and Figure 2: The MODIS-based BRDF reflectance patterns over the ocean need to be better explained. There are two areas of high reflectance, one in the upper right hand part of the figure and another one off the west coast of South America. Given the overpass time of OMI around 1 PM, I assume that only the latter is due to specular reflectance around the glint spot. The high values near the eastern boarder of the swath must be due to the Morel parameterization of diffuse light which depends on chlorophyll content. I am surprised that these values are in a similar range as those near the glint spot and that the pattern doesn't resemble the distribution of chlorophyll in the Atlantic.

P8, L259: "for same" -> "for the same"

Section 5: The discussion of the effects on cloud parameters is very short, especially for the O2-O2 algorithm. How do the results compare with the findings of Lin et al. 2015?

Figure 5: How do you explain the fact that the difference in ECF does not converge to

zero at high ECF?

Figures 5 and 8: The x- and y-axis scales in the left-hand figures should be identical and the 1:1 line should be displayed as reference.

P10, L319: "for unpolluted NO2 mixing ratios" -> "for unpolluted NO2 mixing ratio profiles" (since the profile shape matters, not the absolute NO2 values).

P10, L320: I don't agree with this statement. What is shown here is only to a minor extent a "BRDF effect" (see my major concerns above).

P11, L331: Same issue: It is not correct to state that "BRDF reduces ..".

References:

Kuhlmann, G., Lam, Y. F., Cheung, H. M., Hartl, A., Fung, J. C. H., Chan, P. W., and Wenig, M. O.: Development of a custom OMI NO2 data product for evaluating biases in a regional chemistry transport model, Atmos. Chem. Phys., 15, 5627-5644, doi:10.5194/acp-15-5627-2015, 2015.

Lin, J.-T., Martin, R. V., Boersma, K. F., Sneep, M., Stammes, P., Spurr, R., Wang, P., Van Roozendael, M., Clémer, K., and Irie, H.: Retrieving tropospheric nitrogen dioxide from the Ozone Monitoring Instrument: effects of aerosols, surface reflectance anisotropy, and vertical profile of nitrogen dioxide, Atmos. Chem. Phys., 14, 1441-1461, doi:10.5194/acp-14-1441-2014, 2014.

Lin, J.-T., Liu, M.-Y., Xin, J.-Y., Boersma, K. F., Spurr, R., Martin, R., and Zhang, Q.: Influence of aerosols and surface reflectance on satellite NO2 retrieval: seasonal and spatial characteristics and implications for NOx emission constraints, Atmos. Chem. Phys., 15, 11217-11241, doi:10.5194/acp-15-11217-2015, 2015.

McLinden, C. A., Fioletov, V., Boersma, K. F., Kharol, S. K., Krotkov, N., Lamsal, L., Makar, P. A., Mar-tin, R. V., Veefkind, J. P., and Yang, K.: Improved satellite retrievals of NO2 and SO2 over the Canadian oil sands and comparisons with surface measure-

ments, Atmos. Chem. Phys., 14, 3637-3656, doi:10.5194/acp-14-3637-2014, 2014.

Noguchi, K., Richter, A., Rozanov, V., Rozanov, A., Burrows, J. P., Irie, H., and Kita, K.: Effect of surface BRDF of various land cover types on geostationary observations of tropospheric NO2, Atmos. Meas. Tech., 7, 3497–3508, doi:10.5194/amt-7-3497-2014, 2014. Popp, C., Wang, P., Brunner, D., Stammes, P., Zhou, Y., and Grzegorski, M.: MERIS albedo climatology for FRESCO+ O2 A-band cloud retrieval, Atmos. Meas. Tech., 4, 463-483, doi:10.5194/amt-4-463-2011, 2011.

Russell, A. R., Perring, A. E., Valin, L. C., Bucsela, E. J., Browne, E. C.,Wooldridge, P. J., and Cohen, R. C.: A high spatial resolution retrieval of NO2 column densities from OMI: method and evaluation, Atmos. Chem. Phys., 11, 8543–8554, doi:10.5194/acp-11-8543–2011, 2011.

Zhou, Y., Brunner, D., Spurr, R. J. D., Boersma, K. F., Sneep, M., Popp, C., and Buchmann, B.: Accounting for surface reflectance anisotropy in satellite retrievals of tropospheric NO2, Atmos. Meas. Tech., 3, 1185–1203, doi:10.5194/amt-3-1185- 2010, 2010.

---

## Referee Comment (RC3) · Anonymous Referee #3 · 13 Jun 2016

In their manuscript "Accounting for the effects of surface BRDF on satellite cloud and trace-gas retrievals: A new approach based on geometry-dependent Lambertian-equivalent reflectivity applied to OMI algorithms", A. Vasilkov et al. report on an approach to include surface BRDF effects in OMI NO2 and cloud retrievals. The algorithm is based on the use of MODIS BRDF values for land and a simplified model of surface reflection for the ocean which are then used as geometry dependent LER input parameter for existing lookup-tables of air mass factors and intensities. The algorithm

is applied to real OMI data and the results are compared to those obtained with the standard OMI LER climatology.

The topic of the manuscript is interesting and relevant for UV/vis satellite retrievals of atmospheric parameters where BRDF effects are currently mostly ignored. The approach suggested by the authors is attractive as it would require only small changes to current retrieval schemes and would not much increase computational requirements. The paper is to my knowledge also the first to investigate the effect of BRDF on OMI cloud parameters. The manuscript is well written, clearly structured and contains adequate illustration of the results in figures.

Unfortunately, there are several important shortcomings in the study as outlined below, and I can therefore not recommend the current version of the manuscript for publication in AMT. In my opinion, major revisions are needed before it can be considered again for publication.

**Major comments:**

1. The most important problem with the manuscript is that the "new approach based on geometry-dependent Lambertian-equivalent reflectivity" is – at least as far as I understand – not new but identical to the approach already evaluated by Zhou et al, 2010 and Noguchi et al., 2014 who named it "BRF-approach". Both studies show that this approach is not properly accounting for BRDF effects, which is not surprising as it replaces the direct surface reflectance term with the appropriate value but leads to a wrong source function for the diffuse radiation field. It therefore has a tendency to overestimate BRDF effects.

   In their manuscript, the authors need to discuss previous evaluations of this approach and compare the results of their approximation with those from calculations using the full BRDF treatment. Without such a comparison, it is not clear what the uncertainty of their approximation is.

[Figure]

2. The second problem of the manuscript is that comparisons are made to calculations using OMI LER which is based on a different approach applied to a different data set than the MODIS surface product used in their new algorithm. Therefore, no clear separation of BRDF effects and the effects of other differences between the two products can be made which is an important limitation of the study.

   In my opinion, the authors need to add a comparison to a data product using MODIS surface reflectance but without accounting for BRDF effects in order to be able to quantify BRDF effects. The current comparison is also interesting for users as it indicates how large changes in the OMI products would be, but this is a different question.

3. The role of aerosols is only touched upon in the manuscript, but could be quite important in different parts of the algorithm: in the determination of BRDF parameters in the MODIS product, in the effect of aerosols on cloud parameters when using the new BRDF and in the importance of BRDF on the results. As aerosols increase scattering they will reduce the importance of BRDF effects (see for example the discussion in Noguchi et al., 2014). In the way the algorithm is set up currently (Rayleigh atmosphere), BRDF effects will be overestimated leading to errors in the cloud parameters and air mass factors.

   The effect of aerosols in the different parts of the algorithm has to be discussed and if possible, the uncertainty introduced by overestimation of BRDF effects be quantified.

4. The current manuscript mainly discusses measurements from one single OMI orbit from November 2006 and is therefore based on a very limited data set. Additional data points are shown in Fig. 13 but it is not clear to me from which orbits they are taken. I'm convinced that the effect of BRDF varies with region, season, and viewing geometry, and this needs to be evaluated if one aims at giving meaningful numbers for the uncertainty introduced by ignoring BRDF effects. Also, the approximation made when using geometry dependent LER may introduce different uncertainties depending on geometry and surface type.

In my opinion, significantly more different situations need to be evaluated in more detail to make the numbers derived for the BRDF effects on OMI products meaningful.

**Minor comments**

- The authors use their own O2-O2 cloud algorithm, presumably because this gives them full control of the settings. They state that very good correlation is found for ECF > 0.2 but this of course is not the range of ECFs later discussed. In that sense the difference to current OMI products may be also influenced by the differences between the two implementations of the O2-O2 algorithm.

- Neglecting oceanic foam may be necessary but will lead to an overestimation of BRDF effects over oceans.

- The authors use a vector RTM. It is however not clear to me from the manuscript how polarisation is treated at the surface – can you please provide some details here.

- When introducing BRDF in the cloud product, wouldn't it make sense to also include an approximate treatment of angular dependencies of the reflection from clouds?

- It might be trivial but can BRDF parameters safely be averaged over all MODIS pixels within one OMI scene? Is this a linear problem?

- Is equation 9 used for the figures? If so, isn't that creating a bias in the analysis?

- Which data is shown in Figure 13?

---

## Author Comment (AC1) · 4 Aug 2016

**Response to reviewer #2**

We thank the reviewer for his/her evaluation of our paper and useful comments that helped improve the manuscript. We appreciate reviewer's time and effort in reviewing the manuscript. Below are our responses to each comment. All reviewer's comments are in the standard font while the responses are in the italic font.

On behalf of the authors,

Alexander Vasilkov

Major issues:

There have been several previous studies (e.g. Russel et al., 2011; McLinden et al., 2014; Kuhlmann et al., 2015) comparing NO2 columns retrieved with MODIS-based surface reflectance/albedo with NO2 columns retrieved using climatological Lambertian Equivalent Reflectances (LER) such as the Kleipool et al. (2008) data set.

We added a reference to the paper by Kuhlmann et al. in the introduction. Other references were already given. We also added the following in the introduction:

"Russel et al. (2011) studied the effect of using different surface albedo products on the NO2 columns and found that the impact of the surface albedo can be up to  $\pm 40\%$  for land."

As properly referenced, there have also been several previous studies on the effect of surface reflectance anisotropy (BRDF effects) on NO2 retrievals (Zhou et al., 2010; Noguchi et al., 2014; Lin et al., 2014, 2015).

The novelty of this study as compared to these earlier ones is the treatment of the surface reflectance as a geometry-dependent LER to account for BRDF effects. To be a valuable contribution to the existing body of literature, the publication needs to demonstrate the advantages and limitations of this approach, but it falls short in doing so for two reasons:

1) Instead of analyzing the effects of geometry-dependent versus geometry-independent LER it only compares results based on MODIS reflectance products with results based on OMI-LER from Kleipool et al. (2008). These results are only little influenced by the geometry-dependence of surface reflectance but are dominated by the large differences between MODIS and OMI-based reflectance data sets, an aspect that has been addressed at length in previous studies. For the same reason, the conclusion in the abstract (and conclusions section) that geometry-dependent LERs can change NO2 vertical columns by up to 50% is very misleading as it gives the wrong impression that this is a direct consequence of the geometry-dependence.

In contrast to the present study, Zhou et al. (2010) differentiated between the effects of switching from an OMI climatological LER to a MODIS reflectance product from the effects of

considering BRDF effects versus not considering such effects. They concluded that considering geometry-dependent versus geometry-independent reflectance changes NO2 by mostly well below 20% and that these changes are smaller than those induced by switching from the OMI-based LER of Kleipool et al. (2008) to MODIS.

We do not think that our results including the statement of up to 50% change in NO2 vertical columns are misleading because we clearly state in the abstract and manuscript that the comparisons of retrievals with geometry-dependent LERs are carried out versus those with climatological LERs. Our goal is to document a new global product that will be publically available and could be easily used in the existing operational satellite trace-gas and cloud algorithms. The existing operational algorithms make use of climatological LER products. A question arises how big differences could occur if the climatological LER product would be replaced with the geometry-dependent LER product. We try to answer this practical question in the paper. That is why we are comparing the retrievals based on the geometry-dependent LER with the retrievals based on the geometry-independent climatological LER. The reviewer is right when saying that the differences may be dominated by the large differences between MODIS and OMI-based reflectance data sets. But our main goal is to provide practical results of the comparisons useful for decision-making of developers of the operational algorithms. We think that theoretical results of considering BRDF effects versus not considering such effects have been sufficiently described by Zhou et al. (2010). We would like to note that Lin et al. (2014) used a similar approach in evaluating their NO2 retrievals with full BRDF treatment: they compared their new product with the DOMINO-2 product which makes use of climatological surface LER.

**To clarify this issue we added the following text in the introduction:**

"The main goal of this paper is to document a new global surface reflectivity product that will be publicly available and could be easily used within several existing operational satellite trace-gas and cloud algorithms. We implement the geometry-dependent LERs based on a MODIS BRDF product and use these LERs within OMI cloud and NO2 algorithms. Henceforth, when we refer to geometry-dependent LERs, this refers to a MODIS-based data set. We compare the cloud and NO2 retrievals based on the geometry-dependent LER with the retrievals based on the climatological LER derived from TOMS and OMI measurements. Henceforth, climatological LERs refer to products derived from OMI and TOMS. The differences between those retrievals include both BRDF effects and possible biases between the MODIS and other instrument (OMI and TOMS) reflectance data sets. The existing operational algorithms make use of climatological LER products. By comparing the products retrieved with the geometry-dependent LER with those retrieved with the climatological LER, we address a practical question of how large the differences in various satellite products would be if the climatological LERs are replaced with the geometry-dependent LERs." 2) The proposed approach of using a geometry-dependent LER instead of a full BRDF treatment is very interesting since it has the potential to simplify the retrieval (e.g. existing look-up-table based retrievals could still be used) while accounting for the influence of surface anisotropic reflectance at least to first order.

However, this is still a simplification with respect to a full BRDF treatment since only the TOA radiance is reproduced but the paths of photons reaching the TOA are not exactly the same as in the case of a full BRDF treatment with likely consequences on the vertical sensitivity profile (box AMF profile). The study fails to demonstrate the implications of such a simplification. The effects of such simplifications were addressed by Zhou et al. (2010) which compared a full BRDF treatment with a treatment taking either the MODIS albedo as LER or taking the BRF value for the given illumination and viewing geometry as LER. In both cases, differences from a full BRDF treatment were significant (see their Fig. 10). A similar analysis is needed for the approach proposed here in order to demonstrate both the advantages of a geometry-dependent LER as compared to a geometry-independent LER (Fig. 1 provides some hints) and the limitations with respect to a full BRDF treatment.

We agree. Indeed, the geometry-dependent LER approach provides an exact match of TOA radiances with the full BRDF approach but not the photon path lengths. This simplification can lead to some biases in the calculation of AMFs and thus to biases in the retrieved NO2 vertical columns. Zhou et al. (2010) have estimated the biases. They compared the box AMFs calculated with the full BRDF with that calculated with black-sky albedo and white-sky albedo. According to their Fig. 3 and corresponding text on page 1190, "The effect of surface treatments is most strongly felt near the surface, where the box AMFs differ by up to 10% in this example". A similar order of the difference is found in comparisons of the NO2 vertical columns in Fig. 10 where "Relative differences are smaller than 12% for most of the domain" (see page 1195 of the paper). We consider those differences to be notable but not significant. However, we carried out calculations of NO2 AMF with full BRDF treatment and compared it with that calculated with the corresponding geometry-dependent LER. Differences in AMFs due to different treatment of the surface appear to be small. We added a figure that shows the comparisons.

We added at the beginning of Section 6 the following:

"The geometry-dependent LER approach provides an exact match of TOA radiances with the full BRDF approach but not the photon path lengths. This simplification can lead to some biases in the calculation of AMFs and thus to biases in the retrieved NO2 vertical columns. Zhou et al. (2010) have estimated the biases. They compared the box AMFs and NO2 vertical columns calculated with the full BRDF with that calculated with black-sky albedo and white-sky albedo. According to their data, maximum differences in the box AMFs are up to 10% at the surface and differences in the NO2 vertical columns are smaller than 12%. We carried out calculations of NO2 scattering weights and AMFs with full BRDF treatment and compared them with that calculated with the corresponding geometry-dependent LER. Fig. 11a shows an example of the

altitude dependence of scattering weights calculated with the full BRDF treatment and the geometry-dependent LER. It can be seen that the difference between the scattering weights is small. An AMF difference for this case is 5.6%. Fig. 11b shows a scatter plot of the full BRDF AMFs versus the geometry-dependent LER AMFs calculated for OMI measurements over the eastern US for orbit 12414 of 14 Nov. 2006. Differences in AMFs due to different treatment of the surface are within  $\pm 6\%$  (at 95% confidence interval) and always less than 10%".

Minor points:

Page 2, Line 24: I suggest to include the MERIS based albedo data set of Popp et al. (2011) which is used in the latest FRESCO cloud algorithm and will be considered also for future TROPOMI products.

Thanks. We added this reference.

P5, L152-155: Please explain why O3 and NO2 slant columns are taken from independent OMI algorithms

Our approach is a particular implementation choice that differs from others published in the literature. We added "This is an implementation choice that is designed to minimize potential errors due to cross talk between  $O_3$ ,  $NO_2$  and  $O_2$ - $O_2$  cross sections during the fitting procedure."

P6, L159: It would probably be useful to make clear that you are referring to vertical column densities of O2-O2.

Done.

P7, L211: "over over" -> "over"

Corrected.

P7, Equation (6): This equation appears incomplete as the rightmost term only multiplies two unit-less numbers (reflectance times transmittance) but does not represent a radiance.

Eq. 6 is correct, but the definition of T not straight-forward. We changed the definition of T to clarify that it is in units of radiance, not unitless. To clarify this we changed the definition of T to the following:

"*T* is the total (direct + diffuse) solar irradiance reaching the surface converted to the ideal Lambertian-reflected radiance by diving by pi and multiplied by the transmittance of the reflected radiation in the direction of a satellite instrument."

P8ff: The manuscript structure would probably become clearer by introducing a new section "4 Results and Discussion" and making the present sections 4-6 subsections of this.

We agree. However, to keep the traceability of manuscript changes for all reviewers we decided to preserve the original manuscript structure for now.

P8, L244-245: This is only true over land, not over the ocean.

Correct, added "over land" to make this distinction.

P8, L254ff and Figure 2: The MODIS-based BRDF reflectance patterns over the ocean need to be better explained. There are two areas of high reflectance, one in the upper right hand part of the figure and another one off the west coast of South America. Given the overpass time of OMI around 1 PM, I assume that only the latter is due to specular reflectance around the glint spot. The high values near the eastern boarder of the swath must be due to the Morel parameterization of diffuse light which depends on chlorophyll content. I am surprised that these values are in a similar range as those near the glint spot and that the pattern doesn't resemble the distribution of chlorophyll in the Atlantic.

We agree that more discussion of the angular dependence of geometry-dependent LER over the ocean is needed. We added in Section 4 the following:

"The total ocean reflectance comprises of three components: direct and diffuse solar light reflected from the ocean surface and water-leaving light. A fraction of each component strongly depends on geometry. Reflection of direct solar light dominates in the sun glint area. At the edges of the swath the relative contribution of reflected diffuse light increases because the sky radiance increases to the horizons and the reflection angle increases thus the Fresnel reflection increases. The higher values of LER nearer to the eastern part of the swath than at the western part are mostly due to sky light reflected from the ocean surface. An angular distribution of the sky radiance is not symmetric in the plane of satellite observations because the sun is in the western part of the swath. The sky radiance is higher in the eastern part of the swath and it is reflected at higher angles than the light from the western part of the swath. This is confirmed by our calculations of the view angle dependence of the reflected light only, i.e. no water-leaving radiance included."

P8, L259: "for same" -> "for the same"

Done.

Section 5: The discussion of the effects on cloud parameters is very short, especially for the O2-O2 algorithm. How do the results compare with the findings of Lin et al. 2015?

We agree. We replaced the last sentence in Section 5.2 by the following paragraph:

"The effect of replacing the climatological surface LER by the geometry-dependent LER is much more pronounced for the O2-O2 OCP retrievals than for the RRS retrievals. This can be explained by two physical factors. Firstly, the Rayleigh optical depth of the atmosphere in the UV (the spectral window of the RRS cloud algorithm is 345 - 354 nm) is much higher than in the visible (the wavelength of the O2-O2 OCP retrieval is 477 nm). Higher scattering in the UV leads to a larger fraction of diffuse light illuminating the surface thus decreasing BRDF effects. In the visible, the smoothing effect of Rayleigh scattering is less than in the UV thus resulting in larger BRDF effects. Secondly, sensitivities of the OCP, derived from RRS and O2-O2, to surface reflectivity are different for the RRS and O2-O2 algorithms. The light path of direct sunlight reflected by the surface does not contribute to the RRS signal, because there is no Raman scattering involved. But this direct light path does contribute to O2-O2 absorption. That is why the RRS algorithm is generally less sensitive to the surface and to its BRDF for low cloud fractions. For high surface reflectivity, the reflected direct solar light significantly contributes to TOA radiance therefore causes the OCP differences related to the absence of RRS in direct solar light and the presence of O2-O2 absorption in direct solar light. However, for low surface reflectivity, this mechanism becomes less significant because a fraction of the reflected direct solar light in the TOA radiance is smaller."

We also added the following:

"Lin et al. (2014) compared ECFs and OCPs derived from O2-O2 absorption using the OMI operational algorithm and their own algorithm that makes use of SCDs from the operational algorithm and a set of ancillary parameters that include MODIS BRDF. Their scatter plots of the operational ECF and OCP retrievals versus the new retrievals with BRDF shown in their Fig. 2 are qualitatively similar to ours."

Figure 5: How do you explain the fact that the difference in ECF does not converge to zero at high ECF?

Data in Fig. 5 are for ECF less than 0.25 only (the most interesting range of ECF for trace-gas retrieval). The difference in ECF does converge to zero at high ECF. To clarify this, we added a scatter plot of ECF with climatological LER versus ECF with BRDF for the entire range of ECF. The scatter plot shows that the difference in ECF diminishes with increasing ECF. We also added to the text:

"Figure 5a is a scatter plot of ECF retrieved with the geometry-dependent ECF versus ECF retrieved with climatological LER for the entire range of ECF. It shows that the scatter of data around the 1:1 line diminishes with increasing ECF, i.e. the difference in ECFs decreases with increasing ECF as expected."

Figures 5 and 8: The x- and y-axis scales in the left-hand figures should be identical and the 1:1 line should be displayed as reference.

Corrected. The 1:1 lines are added and mentioned in the captions.

P10, L319: "for unpolluted NO2 mixing ratios" -> "for unpolluted NO2 mixing ratio profiles" (since the profile shape matters, not the absolute NO2 values).

Corrected.

P10, L320: I don't agree with this statement. What is shown here is only to a minor extent a "BRDF effect" (see my major concerns above).

Agree. We clarified here:

"An effect of replacing the climatological LER with geometry-dependent LER ... "

P11, L331: Same issue: It is not correct to state that "BRDF reduces ..".

Agree. We changed this statement:

Replacing the climatological LER with geometry-dependent LER reduces the surface LER ... "

---

## Author Comment (AC2)

Response to reviewer #1

We thank the reviewer for his/her evaluation of our paper and useful comments that helped improve the manuscript. We appreciate reviewer's time and effort in reviewing the manuscript. Below are our responses to each comment. All reviewer's comments are in the standard font while the responses are in the italic font.

On behalf of the authors,

Alexander Vasilkov

Main points

(1)The start of the introduction, from l. 26 – l. 52 around Eqs. 1-3, seems detached from the remainder of the paper. The symbols and terms are different. The text and formulae are unclear. What is the link to the LER, which is used in the remainder of the paper?

*In this section we provide basic definitions of the BRDF and related quantities (BSA and BRF) for informational purposes because there can be different definitions of those quantities in the literature. The BRDF as defined by Eq. 1 is parameterized in Section 3 using a linear combination of three RTLS kernel functions. The coefficients of those functions are then used in radiative transfer computations to calculate the LER with Eq. 6. Indeed, Equations 2 and 3 are not used in our computations. We give those definitions because those quantities are widely used in the literature (see e.g. Zhou et al., 2010; McLinden et al., 2014) and we would like to show a link of our approach to others. We have provided an extra sentence and a break in paragraph 1 of the introduction to make this clearer:*

*"Here, we give some basic definitions that have been used in the literature to provide context to our problem and for clarity as sometimes different definitions have been used for similar or the same quantities."*

- What is I(\omega) in Eq. 1? Is that the same as I_m in Eq. 4? What is the relation to the top-of-atmosphere radiance as observed by OMI ?

*$I(\omega_r)$ is not the same as $I_m$ in Eq. 4. $I_m$ is the top-of-atmosphere radiance as observed by OMI while $I(\omega_r)$ is the reflected radiance in the direction $\omega_r$ at the surface. $I(\omega_r)$ provides a boundary condition at the surface for calculation of the TOA radiance. We added in the text:*

*"$I(\theta_i)$ is the radiance incident on the surface" and*

*"The reflected radiance $I(\theta_r)$ is calculated by integrating the product of BRDF and dF over all directions of the incident radiation. $I(\theta_r)$ provides a boundary condition at the surface for computations of the top-of-atmosphere radiance."*

Why do you use solid angle \omega, whereas in the remainder of the paper you use \theta and \phi? Explain \theta_r, which is called \theta in the remainder of the paper.

*We use the solid angles to (1) simplify the equations and (2) follow the convention in the definition of BRDF. The denotations $\theta_i$ and $\theta_r$ are used in Eq. (1)-(3) only to distinguish the zenith angles of incident and reflected light. $\theta_r$ is the zenith angle of reflected light; the subscript "r" is omitted in the remainder of the paper for simplicity. We clarify that by adding the following immediately after Eq.2:*

*"where $\theta_r$ is the zenith angle of reflected light (subscript "r" is omitted in the reminder of the paper for simplicity)"*

Why is F, mentioned below Eq. 1, not used in the equation?

*We introduced the quantity F into Eq. 1, adding one more equality:*

*"BRDF=dI/dF=…"*

- In Eq. 2 please give the integration limits. Below Eq. 2 it is apparently assumed that this particular Lambertian has an albedo of 1. But also for a less reflective Lambertian surface the relation can be used.

*We added "where integration is carried out over the solid angle of $2\pi$ for the upper hemisphere". Eq. 2 gives a general definition. The sentence below Eq. 2 considers a particular case of the perfect Lambertian surface that has albedo of 1. We have clarified this in a revised paper.*

- Eq. 3: does BRF in this equation yields Rg in Eq. 4?

*Yes, if a value of BRF is used as $R_g$ in Eq. 4. In general, $R_g$ can be a geometry-dependent or climatological LER.*

(2) The interpretation of the scatter plots of retrieved cloud parameters from RRS and O2-O2 algorithms between the BRDF-dependent LER and climatology LER, especially the OCP, deserves more discussion. See Figs. 7 and 9. Apparently the RRS OCP is hardly depending on the surface BRDF, whereas the O2-O2 OCP is strongly depending on it. That is remarkable. It cannot be only explained by the decrease of Rayleigh scattering at 466 nm as compared to 354 nm, as stated in the paper. Another difference in both algorithms must be causing this. It is probably due to the fact that the RRS signal is not including all light paths that are relevant for the O2-O2 absorption (and NO2 absorption). Namely, the direct light path of direct sunlight reflected by the surface and arriving at the satellite is not included in the RRS signal, because there is no Rayleigh (Raman) scattering involved. But it is an important light path for the cloud-free part of the pixel. And this direct light path is also strongly contributing to O2-O2 (and NO2) absorption. So the RRS method is in first order insensitive to the surface and to its BRDF. Only

via the light paths Rayleigh + surface reflection, and surface reflection + Rayleigh can the RRS signal pick up surface BRDF effects. But that is a second-order effect. Please consider this cause in explaining the OCP behaviour of the RRS and O2-O2 algorithms.

*We particularly thank the reviewer for this comment. We agree that the differences between the RRS and O2-O2 cloud algorithms deserve more discussion. Indeed, sensitivities of the OCP, derived from RRS and O2-O2, to surface reflectivity are different for the RRS and O2-O2 algorithms. This is because RRS first decreases with increase in surface reflectivity, and then it starts to slowly increase, while O2-O2 absorption increases monotonically. Also, the sensitivity to surface reflectivity depends upon the reflectivity itself. For high surface reflectivity, the reflected direct solar light significantly contributes to TOA radiance, and therefore causes the OCP differences related to the absence of RRS in direct solar light and the presence of O2-O2 absorption in direct solar light. However, for low surface reflectivity, this mechanism becomes less significant.*

*We replaced the last sentence in Section 5.2 by the following paragraph:*

*"The effect of replacing the climatological surface LER by the geometry-dependent LER is remarkably more pronounced for the O2-O2 OCP retrievals than for the RRS retrievals. This can be explained by two physical factors. Firstly, the Rayleigh optical depth of the atmosphere in the UV (the spectral window of the RRS cloud algorithm is 345 - 354 nm) is much higher than in the visible (the wavelength of the O2-O2 OCP retrieval is 477 nm). Higher scattering in the UV leads to a larger fraction of diffuse light illuminating the surface thus decreasing BRDF effects. In the visible, the smoothing effect of Rayleigh scattering is less than in the UV thus resulting in larger BRDF effects. Secondly, sensitivities of the OCP, derived from RRS and O2-O2, to surface reflectivity are different for the RRS and O2-O2 algorithms. The direct light path of direct sunlight reflected by the surface does not contribute to the RRS signal, because there is no Raman scattering involved. But this direct light path does contribute to O2-O2 absorption. That is why the RRS algorithm is generally less sensitive to the surface and to its BRDF for low cloud fractions. For high surface reflectivity, the reflected direct solar light significantly contributes to TOA radiance therefore causes the OCP differences related to the absence of RRS in direct solar light and the presence of O2-O2 absorption in direct solar light. However, for low surface reflectivity, this mechanism becomes less significant because the fraction of the reflected direct solar light in the TOA radiance is smaller."*

(3) Please add histograms of ECF and OCP for the orbits shown, and not only scatter plots, to see the difference between including and excluding BRDF effects, and the difference between RRS and O2-O2 algorithms.

*We added histograms of ECF and OCP in Figs. 5, 7, 8, and 9. We also added the following in Section 5.1:*

*"Figure 5d shows normalized histograms of ECFs for 0.05<ECF<0.25. The normalized histograms of ECF retrieved with climatological LER and ECF retrieved with BRDF are close to each other. This reflects small differences between the ECFs on average."*

*and the following in Section 5.2:*

*"The histograms of OCP retrieved from the O2-O2 cloud algorithm (Fig. 9c) noticeably differ from that retrieved from the RRS cloud algorithm (Fig. 7c). According to Fig. 9c, lower altitude clouds (with OCP > 800 hPa) are observed more frequently over the ocean than over land. For high altitude clouds (OCP < 450 hPa) the situation is reverse: they observed more frequently over land than over the ocean. Both patterns in the vertical distribution of clouds are much less pronounced in the histograms of OCP retrieved from the RRS algorithm."*

Questions and textual comments

- Eq. 4: please say that I_g and I_c are at top-of-atmosphere

*Done*

- L. 67: is Ac=0.8 also assumed in this paper?

*Yes. We added: "In this paper we also assume R_c=0.8 for the OMI cloud and NO2 algorithms".*

- L. 71: add here a reference to Stammes et al. (2008)

*Thanks, done.*

- L. 76: add here a reference to Sneep et al. (2008)

*Done.*

- L. 121: remove: its

*Corrected.*

- L. 141: does the RRS ECF hold for Rc=0.8 ?

*Yes, see the comment to L. 67 above.*

- L. 175: please make a separate equation of the in-text formula.

*Done.*

- L. 185: please clarify: do you use in the paper the climatological ratio Rg(354)/Rg(470) or a ratio of unity?

*We use the ratio of unity in the paper because (1) the climatological ratio Rg(354)/Rg(470) can be close to unity for some types of land. An example of the spectral dependence of climatological LER is shown in Figure below. (2 )we want to avoid possible uncertainties that could be potentially involved with the use of climatological spectral dependence of LER from existing data sets. The possible uncertainties are related to inconsistency of the spectral LERs from different data sets. For instance, according to the climatological data base of Kleipool et al. (2008), land is brighter in the UV than in the VIS for most areas (see Fig. 15 of Kleipool's paper), which contradicts the common understanding that the land is darker in the UV than in the VIS. We rewrote the text:*

*"In the paper we assume that the BRDF coefficents are spectrally independent to focus on the surface BRDF effects only. Using climatological data of Kleipool et al. (2008) we find that this assumption can be valid for some areas, e.g. the climatological ratio Rg(354)/Rg(470) is close to unity (within ±5%) over the eastern part of North America. However, this is not the case for arid and semi-arid areas. We plan to release our geometry-dependent LER product computed for wavelengths other than 470 nm using a spectral correction of the BRDF coefficients. This spectral correction will be based on the ratio Rg(354)/Rg(470) derived from a critical analysis of different existing data sets of climatological satellite-derived LERs."*

[Figure]

*Figure. Spectral dependence of the climatological OMI-derived surface reflectivity over North America along the longitude of $82^0$ for different latitudes ($30^0$ to $42^0$ N).*

- L. 208: please give a reference for MYD43GF.

*Typo, should be MCD43GF. We additionally provided a link to the product: ftp://rsftp.eeos.umb.edu/data02/Gapfilled/.*

- L. 215: I_TOA: why is a new symbol introduced? Where the other radiance symbols not at top-of-atmosphere? How does it relate to I_m of Eq. 4?

*We introduced a new symbol $I_{TOA}$ for the computed TOA radiance to distinguish it from the measured TOA radiance $I_m$. To avoid a possible confusion, we replaced $I_{TOA}$ with a new symbol $I_{comp}$.*

Please do not introduce unnecessarily new quantities and symbols. Please also relate \theta, \phi, and \theta_0 to the earlier introduced angles.

*The angles $\theta$ and $\varphi$ characterize the observational geometry at TOA; the angle $\theta_0$ is the SZA. The earlier introduced angles in Eq. 1 and 2 are defined at the surface.*

- L. 217 ff: the explanation of T is unclear. T is the total two-way transmission of the atmosphere.

*The explanation of T is not simple (ours adopted from the original paper by Dave, 1978). T is not simply the total two-way transmission of the atmosphere because transmission is dimensionless while T has the dimension of radiance. We modified the definition of T to the following:*

*"T is the total (direct + diffuse) solar irradiance reaching the surface converted to the Lambertian-reflected radiance by diving by $\pi$ and multiplied by the transmittance of reflected radiation between the surface and TOA in the direction of a satellite instrument."*

- L. 245: Land is mostly darker in the UV than in the VIS. So why not use the climatological OMI data base at 354 nm?

*According to Fig. 15 of Kleipool et al. (2008), land is brighter in the UV than in Vis for grasses, broadleaf forests, and other types. See the answer to comment to L. 185.*

- L. 265: please indicate the orbit and date.

*Done.*

- L. 300: is in Sect. 6 only the O2-O2 algorithm used and not the RRS algorithm because the latter has very little impact of BRDF?

*The O2-O2 cloud product is used in Section 6 because the wavelengths which it uses (466 for ECF and 477 nm for OCP) are closer to the NO2 fitting window 405-465 nm  specified in Section 2.3.2. , and therefore light at these wavelengths follows a more similar light path compared to those the RRS algorithm uses, in the UV.*

- L. 301: why are the NO2 profile shapes from June and not from November, for which month the satellite data were chosen?

*We thank the reviewer for pointing out this inconsistency. We have redone the figure using the November profiles.*

- L. 313: can you please explain how this formula is derived?

*Eq. 8 was derived by differentiating Eq.7 assuming that AMFg and fr are independent variables and both depend on delta(Rg). We clarified this in the manuscript.*

- L. 355: please mention here that the background aerosols are included in the climatological LER, but are missing in the BRDF, so that the ECF from the BRDF has a low bias.

*We added the following:*
*"It should be noted that the background aerosols are included in the climatological LER; therefore, they are virtually accounted for in the ECF derived using the LER climatology. The geometry-dependent LER is calculated for aerosol-free conditions, thus the corresponding ECF should have a bias."*

- L. 366: the use > to use

*Done.*

- L. 385: missing: Chandrasekhar

*Done.*

Figures and captions:

Fig. 1: please use larger font for lat/lon (like in Fig. 2). What is the spatial resolution of these maps?

*Done, font size was increased. The spatial resolution of the maps is equal to the original resolution of the MODIS-derived BRDF product, i.e. 30 arc sec or about 1 km. We added this information to the figure caption.*

Fig. 2: please use a, b, c for the subplots. This also holds for the other figures with 3 subplots.

*Done.*

Fig. 4: which orbit and date? With which LER figure should this be compared? RRS-derived > RRS-retrieved

*Orbit 12414 of 14 Nov 2006. Data in Fig. 4 correspond to LER shown in Fig. 3 (b). We added this to the caption. Corrected.*

FIg. 5: which orbit and date?

*Orbit 12414 of 14 Nov 2006, now listed in the caption.*

Fig. 7: please write out the caption.

*Done.*

For Fig. 7 and Fig. 9 please consider inverting the axes, thus from 1000 to 0 hPa, because that looks more natural (low clouds at the origin of the plot).

*Done.*

Fig. 9: Please use better caption; the reference to the caption of Fig. 5 leads to another reference to another caption. What are the straight lines in the left plot?

*Changed to*

*"(a) Scatter plot of O2-O2-retrieved OCPs computed with geometry-dependent LERs versus climatological LERs, the 1:1 line is in black; (b) similar for ECF< 0.25 with linear fits; (c) the mean ECF difference (diamonds) and standard deviation (error bars) as a function of ECF; (d) normalized histograms of OCP."*

Fig. 10: reflectivity > surface reflectivity

*Done.*

Fig. 11: which orbit and date?

*Orbit 12414 of 14 Nov 2006, now listed in the caption.*

Fig. 12: please number the subplots. Please add LER, OCP, fr to the legend of the lower 3 subplots.

*Done.*

Fig. 13: which date and orbit?

*Orbit 12414 of 14 Nov 2006 for data over America and orbit 12391 of 13 Nov 2006 for data over China, now listed in the caption.*

---

## Author Comment (AC3)

Response to reviewer #3

We thank the reviewer for his/her evaluation of our paper and useful comments that helped improve the manuscript. We appreciate reviewer's time and effort in reviewing the manuscript. Below are our responses to each comment. All reviewer's comments are in the standard font while the responses are in the italic font.

On behalf of the authors,

Alexander Vasilkov

Major comments:

1. The most important problem with the manuscript is that the "new approach based on geometry-dependent Lambertian-equivalent reflectivity" is – at least as far as I understand – not new but identical to the approach already evaluated by Zhou et al, 2010 and Noguchi et al., 2014 who named it "BRF-approach". Both studies show that this approach is not properly accounting for BRDF effects, which is not surprising as it replaces the direct surface reflectance term with the appropriate value but leads to a wrong source function for the diffuse radiation field. It therefore has a tendency to overestimate BRDF effects.

*Our approach is not identical to the approach already evaluated by Zhou et al, 2010 and Noguchi et al., 2014. Firstly, we have developed a product that could be easily used in the existing operational satellite trace-gas and cloud algorithms based on the MLER concept. The use of this product in the existing algorithms does not require extensive computational efforts. The "BRF approach" cannot be easily used in the existing operational satellite trace-gas and cloud algorithm. Moreover, it requires extensive radiative transfer (RT) computations that prevent from the use of a vector RT code which is necessary. Next, our product is global: it is applicable to the ocean unlike Zhou et al, 2010 and Noguchi et al., 2014.*

In their manuscript, the authors need to discuss previous evaluations of this approach and compare the results of their approximation with those from calculations using the full BRDF treatment. Without such a comparison, it is not clear what the uncertainty of their approximation is.

*We carried out calculations of NO2 AMF with full BRDF treatment and compared it with that calculated with the corresponding geometry-dependent LER. We added a figure that shows the comparisons. We also added at the beginning of Section 6 the following:*

*"The geometry-dependent LER approach provides an exact match of TOA radiances with the full BRDF approach but not the photon path lengths. This simplification can lead to some biases in the calculation of AMFs and thus to biases in the retrieved NO2 vertical columns. Zhou et al. (2010) have estimated the biases. They compared the box AMFs and NO2 vertical columns calculated with the full BRDF with that calculated with black-sky albedo and white-sky albedo.*

*According to their data, maximum differences in the box AMFs are up to 10% at the surface and differences in the NO2 vertical columns are smaller than 12%. We carried out calculations of NO2 scattering weights and AMFs with full BRDF treatment and compared them with that calculated with the corresponding geometry-dependent LER. Fig. 11a shows an example of the altitude dependence of scattering weights calculated with the full BRDF treatment and the geometry-dependent LER. It can be seen that the difference between the scattering weights is small. An AMF difference for this case is 5.6%. Fig. 11b shows a scatter plot of the full BRDF AMFs versus the geometry-dependent LER AMFs calculated for OMI measurements over the eastern US for orbit 12414 of 14 Nov. 2006. Differences in AMFs due to different treatment of the surface are within $\pm 6\%$ (at 95% confidence interval) and always less than 10%".*

2. The second problem of the manuscript is that comparisons are made to calculations using OMI LER which is based on a different approach applied to a different data set than the MODIS surface product used in their new algorithm. Therefore, no clear separation of BRDF effects and the effects of other differences between the two products can be made which is an important limitation of the study.

*Our goal is to document a new global product that will be publically available and could be easily used in the existing operational satellite trace-gas and cloud algorithms. The existing operational algorithms make use of climatological LER products. A question arises how big differences could occur if the climatological LER product would be replaced with the geometry-dependent LER product. We try to answer this practical question in the paper. That is why we are comparing the retrievals based on the geometry-dependent LER with the retrievals based on the geometry-independent climatological LER. The reviewer is right when saying that the differences may be dominated by the large differences between MODIS and OMI-based reflectance data sets. But we are aimed to obtain practical results of the comparisons useful for decision-making of developers of the operational algorithms. We think that theoretical results of considering BRDF effects versus not considering such effects have been sufficiently described by Zhou et al. (2010).*

*To clarify this issue we have made the following additions to the introduction:*

*"The main goal of this paper is to document a new global surface reflectivity product that will be publicly available and could be easily used within several existing operational satellite trace-gas and cloud algorithms. We implement the geometry-dependent LERs based on a MODIS BRDF product and use these LERs within OMI cloud and NO2 algorithms. Henceforth, when we refer to geometry-dependent LERs, this refers to a MODIS-based data set. We compare the cloud and NO2 retrievals based on the geometry-dependent LER with the retrievals based on the climatological LER derived from TOMS and OMI measurements. Henceforth, climatological LERs refer to products derived from OMI and TOMS. The differences between those retrievals include both BRDF effects and possible biases between the MODIS and other instrument (OMI and TOMS) reflectance data sets. The existing operational algorithms make use of*

*climatological LER products. By comparing the products retrieved with the geometry-dependent LER with those retrieved with the climatological LER, we address a practical question of how large the differences in various satellite products would be if the climatological LERs are replaced with the geometry-dependent LERs."*

In my opinion, the authors need to add a comparison to a data product using MODIS surface reflectance but without accounting for BRDF effects in order to be able to quantify BRDF effects. The current comparison is also interesting for users as it indicates how large changes in the OMI products would be, but this is a different question.

*We think that this is a pure theoretical issue, which was sufficiently investigated by Zhou et al. (2010). They have estimated possible NO2 differences due to not accounting for full BRDF. They compared the box AMFs calculated with the full BRDF with that calculated with black-sky albedo and white-sky albedo. According to their Fig. 3 and corresponding text on page 1190, "The effect of surface treatments is most strongly felt near the surface, where the box AMFs differ by up to 10% in this example". A similar order of the difference is found in comparisons of the NO2 vertical columns in Fig. 10 where "Relative differences are smaller than 12% for most of the domain" (see page 1195 of the paper). We mentioned their study in Section 6 (see our answer to the previous comment). We would like to note that both black-sky albedo and white-sky albedo derived from MODIS do not adequately describe the real surface albedo that depends on an exact angular distribution of the sky radiance. That is why we consider such comparisons to be purely theoretical.*

3. The role of aerosols is only touched upon in the manuscript, but could be quite important in different parts of the algorithm: in the determination of BRDF parameters in the MODIS product, in the effect of aerosols on cloud parameters when using the new BRDF and in the importance of BRDF on the results. As aerosols increase scattering they will reduce the importance of BRDF effects (see for example the discussion in Noguchi et al., 2014). In the way the algorithm is set up currently (Rayleigh atmosphere), BRDF effects will be overestimated leading to errors in the cloud parameters and air mass factors.

The effect of aerosols in the different parts of the algorithm has to be discussed and if possible, the uncertainty introduced by overestimation of BRDF effects be quantified.

*The reviewer is absolutely right that the role of aerosols is quite important and the aerosols can reduce the BRDF effects by increasing the diffuse solar light at the surface. We think that their role was carefully studied in several papers, e.g. Lin et al. (2014 & 2015). That is why we intentionally limited our discussion of the aerosol effects. We briefly discussed the aerosol effects in Introduction. We also stated in Section 2.3.1 that our cloud algorithms implicitly accounts for non-absorbing aerosols, treating them as clouds and this increases effective cloud fraction. The aerosol effect is thought to be significant in trace gas algorithms because the aerosol affects*

*AMFs. We mentioned in Conclusions that we plan to explicitly include aerosols in the NO2 algorithm in the future work.*

*We added at the end of Section 3 the following:*

*"It should be noted that aerosols are not included in the computation of the geometry-dependent LER. Scattering by aerosols in the atmosphere reduces the BRDF effects (Noguchi et al., 2014). Therefore, the use of the geometry-dependent LER may result in overestimation of the BRDF effects. While non-absorbing aerosols are implicitly accounted for in the cloud algorithms (see Section 2.3.1), the aerosols directly affect the Air Mass Factor (AMF), thus trace gas retrievals."*

*and the following in Section 2.3.1:*

*"However, the increase of cloud fraction due to the presence of aerosols cannot correctly reproduce an increase of diffuse solar light at the surface caused by aerosol scattering. This may introduce some error in calculation of the clear-sky subpixel radiance because the BRDF effect depends on a ratio of diffuse to direct solar light."*

4. The current manuscript mainly discusses measurements from one single OMI orbit from November 2006 and is therefore based on a very limited data set. Additional data points are shown in Fig. 13 but it is not clear to me from which orbits they are taken. I'm convinced that the effect of BRDF varies with region, season, and viewing geometry, and this needs to be evaluated if one aims at giving meaningful numbers for the uncertainty introduced by ignoring BRDF effects. Also, the approximation made when using geometry dependent LER may introduce different uncertainties depending on geometry and surface type.

In my opinion, significantly more different situations need to be evaluated in more detail to make the numbers derived for the BRDF effects on OMI products meaningful.

*To present our results, we selected OMI orbit 12414 because it contains land and ocean areas in approximately equal proportions. Data in Fig. 13 are for orbits 12391 and 12414. We agree that more data from different conditions and seasons need to be evaluated to make the numbers more representative. To look at BRDF variations with region and viewing geometry we process OMI data for the entire day of Nov 14, 2006. To evaluate BRDF variations with season we processed OMI data for one more day in summer (July 14, 2006). We added a figure that shows the ECF and OCP retrievals from the O2-O2 cloud algorithm for those two days. We also added the following text in Section 5:*

*"To make the numbers characterizing the ECF and OCP differences be more representative, we processed OMI data for two days of November 14 and July 14, 2006. Figure 10 shows the ECF and OCP differences as a function of ECF for those two days. The ECF differences calculated for the entire day of Nov 14, 2006 (Fig.10c) are quite close to those calculated for a single orbit 12414 of that day (Fig.8b). The OCP differences over land calculated for the entire day*

*(Fig.10d) are slightly lower than those calculated for orbit 12414 of that day (Fig. 9b) while the OCP differences over ocean for the entire day are quite close to those calculated for one orbit. The ECF and OCP differences are similar for different seasons. A small increase of the OCP differences in November may not be statistically significant. The data in Fig. 10 indicate that the ECF and OCP differences obtained for OMI orbit 12414 are globally representative. "*

*We also calculated the tropospheric NO2 AMFs for two days: Nov 14 and Jul 14, 2006. A global analysis of the AMF differences due to replacing the climatological LER with the geometry-dependent LER shows that the AMF differences for OMI orbit 12414 are quite representative for both days. A figure below shows a global map of the trop AMF and the AMF differences for Nov 14, 2006. We decided not to include the figure in the manuscript but added at the end of Section 6 the following:*

*"To make the numbers characterizing the AMF differences be more representative, we calculated the tropospheric NO2 AMFs using the geometry-dependent LER and compared them with those calculated with the climatological LER for two days: November 14 and July 14, 2006. The AMF differences arising from both replacing the climatological LER with the geometry-dependent LER and changing the cloud parameters exhibit strong spatial variations with smaller effects over the ocean, unpolluted, or cloudy areas similar to Fig. 13. A global analysis of the AMF differences shows that the AMF differences for OMI orbit 12414 are consistent with those for both days."*

[Figure]

Figure. (a) AMF calculated with the geometry-dependent LER; (b) AMF differences.

Minor comments

• The authors use their own O2-O2 cloud algorithm, presumably because this gives them full control of the settings. They state that very good correlation is found for ECF > 0.2 but this of course is not the range of ECFs later discussed. In that sense the difference to current OMI products may be also influenced by the differences between the two implementations of the O2-O2 algorithm.

*The reviewer is right; we use our own O2-O2 cloud algorithm to get full control of the settings. We briefly mention comparisons of our algorithm with the operational O2-O2 algorithm just to provide some information about verification of our own algorithm. We do not use the operational O2-O2 algorithm products in our comparisons. That is why our comparisons are not influenced by the differences between the two implementations of the O2-O2 algorithm.*

• Neglecting oceanic foam may be necessary but will lead to an overestimation of BRDF effects over oceans.

*We added the following in Section 3:*

 *"We neglect contributions from oceanic foam that can be significant for high wind speeds."*

*and Conclusions:*

*" We plan to improve the oceanic model of BRDF by including a variable wind speed and oceanic foam with areal fraction that depends on the wind speed in our computations."*

• The authors use a vector RTM. It is however not clear to me from the manuscript how polarisation is treated at the surface – can you please provide some details here.

*We added to Section 2.1 the following:*

*"We account for polarization at the ocean surface using a full Fresnel reflection matrix as suggested by Mishchenko and Travis (1997)."*

*Mishchenko, M. I. and Travis, L. D.: Satellite retrieval of aerosol properties over the ocean using polarization as well as intensity of reflected sunlight, J. Geophys. Res., 102, 16989–17013, doi:10.1029/96JD02425, 1997.*

• When introducing BRDF in the cloud product, wouldn't it make sense to also include an approximate treatment of angular dependencies of the reflection from clouds?

*All our algorithms are based on the MLER approach, i.e. clouds are treated as an opaque Lambertian surface. That is why we do not consider angular dependencies of the reflection from clouds.*

• It might be trivial but can BRDF parameters safely be averaged over all MODIS pixels within one OMI scene? Is this a linear problem?

*To address to this question we added at the end of Section 3 the following:*

*"Averaging the BRDF coefficients over an OMI pixel may not be equivalent to averaging the high resolution surface LER over the OMI pixel. We carried out a numerical experiment of calculations of TOA radiances using the high resolution BRDF coefficients and OMI geometries for the US Washington-Baltimore corridor area (Fig. 1). The TOA radiances were converted into LERs using Eq. 6 and then the LERs were averaged over OMI pixels. The resulting LERs were compared with that calculated from the standard procedure of averaging the BRDF coefficients first. We found that the mean LER difference was equal to $0.75*10^{-5}$ with the standard deviation of $4.2*10^{-4}$ which is quite acceptable for our purposes."*

• Is equation 9 used for the figures? If so, isn't that creating a bias in the analysis?

*Eq. 9 is not used for the figures. Eq. 9 is mostly intended to illustrate the effect of changing surface reflectance on AMF in cloudy conditions. We clarify this in the manuscript by adding the following:*

*"It should be noted that we derived Eq. 9 and 10 to qualitatively illustrate the effect of changing surface reflectance on AMF in cloudy conditions. The equations are not used to produce data in the figures. The data in the figures of Section 6 are obtained numerically using Eq. 8."*

• Which data is shown in Figure 13?

*Orbit 12414 of 14 Nov 2006 for data over America and orbit 12391 of 13 Nov 2006 for data over China. We added this information to the figure caption.*